# Adaptations of the axon initial segment in fast-spiking interneurons of the human neocortex support low action potential thresholds

Emoke Bakos[1,2☉], Ádám Tiszlavicz[1☉], Viktor Szegedi[1,2],
Abdennour Douida[1], Szabina Furdan [1], Daphne K. Welter[3], Jonathan M. Landry[3],
Balazs Bende[4], Gabor Hutoczki[5], Pal Barzo[6], Gabor Tamas[7], Vladimir Benes[3],
Attila Szucs[8], Karri Lamsa [1,2]*

1 Hungarian Center of Excellence for Molecular Medicine Research Group for Human Neuron Physiology and Therapy, Szeged, Hungary, 2 Department of Physiology, Anatomy and Neuroscience, University of Szeged, Szeged, Hungary, 3 European Molecular Biology Laboratory, Heidelberg, Germany, 4 Hungarian Centre of Excellence for Molecular Medicine Research Group for Translational Medicine Development, Szeged, Hungary, 5 Department of Neurosurgery, University of Debrecen Clinical Centre, Debrecen Hungary, 6 Department of Neurosurgery, University of Szeged, Szeged, Hungary, 7 ELKH-SZTE Research Group for Cortical Microcircuits, Department of Physiology, Anatomy and Neuroscience, University of Szeged, Szeged, Hungary, 8 Neuronal Cell Biology Research Group, Eötvös Loránd University, Budapest, Hungary

☉ These authors contributed equally to this work.
* klamsa@bio.u-szeged.hu, karri.lamsa@hcemm.eu

## Abstract

The mammalian brain exhibits notable interspecies variation. Microanatomical and molecular differences in homologous neurons, those with similar locations and developmental origins across species, are best characterized in the neocortical mantle, the center of complex brain functions; however, the purpose of these differences remains unclear. We performed whole-cell microelectrode recordings along with microanatomical and molecular analyses of human fast-spiking parvalbumin (pvalb)-expressing interneurons in neocortical tissue resected during brain surgery, comparing them with similar data obtained from the mouse neocortex. The action potential (AP) firing threshold was lower in human neurons than in mouse neurons. This was due to a deficiency in low-voltage–activated inhibitory Kv1.1 and Kv1.2 potassium channels in the axon initial segment (AIS), a specialized axonal region that determines AP threshold and initiation, in human cells. In contrast, Kv1 ion channels were prominent in mouse neurons. The AIS was also moderately elongated in humans. Computational simulations of fast-spiking interneurons revealed that the human-type AIS lowers the AP threshold and shortens the time lag for AP initiation. We found that the low membrane AP firing threshold in pvalb neurons is closely linked to slow membrane potential kinetics in the soma. Thus, the human AIS supports fast in–fast out circuit function in human pvalb neurons, compensating for electrically slow somatic membrane responses. When formulating therapeutic strategies that involve fast-spiking neurons, it is crucial to take into account the molecular and functional species differences.

**Data availability statement:** All relevant data are within the paper and its Supporting information files.

**Funding:** Project no. TKP-2021-EGA-05 was implemented with support provided by the Ministry of Culture and Innovation of Hungary from the National Research, Development, and Innovation Fund (https://nkfih.gov.hu), financed under the TKP2021-EGA funding scheme (K.L.). Project no. 2022-2.1.1-NL-2022-00005 was implemented with support provided by the Ministry of Culture and Innovation of Hungary from the National Research, Development, and Innovation Fund (https://nkfih.gov.hu), financed under the 2022-2.1.1-NL funding scheme (K.L.). Additional funding was received from the EU's Horizon 2020 research and innovation program (https://ec.europa.eu/programmes/horizon2020) under grant agreement no. 739593 (K.L.);Nemzeti Kutatási, Fejlesztési és Innovaciós Alap OTKA K 134279 and ADVANCED_150382 (https://nkfih.gov.hu) (K.L.); the Hungarian Scientific Research Foundation (https://nkfih.gov.hu) grant number ANN-135291 (K.L.); Eötvös Loránd Research Network (https://hun-ren.hu) grants ELKH-SZTE Agykérgi Neuronhálózatok Kutatócsoport and KÖ-36/2021 (G.T.); Ministry of Human Capacities Hungary (20391-3/2018/FEKUSTRAT (G.T.) and NKP 16-3-VIII-3) (G.T.); National Research, Development and Innovation Office (https://nkfih.gov.hu) grants GINOP 2.3.2-15-2016-00018 (G.T.); Élvonal KKP 133807 (G.T.), ÚNKP-20-5 - SZTE-681 (G.T.), 2019-2.1.7-ERA-NET-2022-00038 (G.T.), TKP2021-EGA-09 (G.T.), TKP-2021-EGA-28 (G.T.), and OTKA K128863 (G.T.). In addition, funding was received from National Institutes of Health (https://www.nih.gov) awards U01MH114812 (G.T.) and UM1MH130981 (G.T.); and National Research, Development, and Innovation Office (https://nkfih.gov.hu) grant no. ANN_135291 (A.S.). Grant No. 8110 was received from the University of Szeged for the open access publication of the manuscript (https://szerzoknek.ek.szte.hu/tamogatott-oa-publikalas/). The funders had no role in study design, data collection and analysis, decision to publish, or preparation of the manuscript.

**Competing interests:** The authors have declared that no competing interests exist.

**Abbreviations:** AIS, axon initial segment; AP, action potential; AP AHP, action potential after-hyperpolarization; IQR, interquartile range.

## Author summary

Fast-spiking neurons in the human neocortex feature molecular specializations in the axon initial segment that lower firing thresholds and minimize input–output delay.

## Introduction

All organs in the body evolved specialized functions to optimally support animal life and vitality. The mammalian brain exhibits interspecies differences not only in volume and cortical convolution but also in nerve cell microstructure and molecular composition [1,2]. Although the brain comprises homologous neurons, sharing similar locations and developmental origins across species, these cells show species-specific differentiation at the microscale level [3–6]. The purpose of this neuronal specialization across species remains poorly understood [7,8], although these adaptations likely optimize neural circuit function [9]. Neuron types and their species-specific features are best characterized in the neocortical mantle, the center of complex high-order brain functions [1,5,6,10–19]. Neocortical function has been intensively studied in relation to brain cognitive and behavioral processes in humans and animal models, the latter aiding in elucidating the roles of neuron types in these processes [20–22]. However, studies using resected human brain tissue have revealed various "human-specific" transcriptomic and microstructural features in neocortical neurons [7,10,11,23–31]. Many of these differences likely contribute to species-specific neuronal functions; however, the structure–function relationship remains unclear [12,16,17,19,30,32–34].

Distinctive differences in homologous neurons between the rodent and human neocortex include electrical resistivity and the membrane potential level required to trigger action potentials (APs) [25]. Human neocortical excitatory pyramidal neurons exhibit lower input resistance (Rin) and require greater membrane depolarization (i.e., a higher AP firing threshold) to trigger APs compared with rodent neurons [24,29,35]. In contrast, inhibitory interneurons in the human neocortex display higher input resistance and a lower AP firing threshold relative to their rodent counterparts [26,27]. Prior studies suggest that these interspecies differences also apply to fast-spiking interneurons [28,31,36], the principal inhibitory neuron population in the neocortex, characterized by the expression of *parvalbumin* (*PVALB*), the gene encoding pvalb protein [23]. In humans, pvalb neurons exhibit elevated input resistance; therefore, they generate membrane potential ($V_m$) changes with reduced electric input [28]. For the same reason, $V_m$ changes, such as excitatory synaptic potentials that lead to APs, occur more slowly [28,31,36–39]. The delayed $V_m$ response in human fast-spiking neurons, compared with that in their rodent counterparts, appears counterintuitive, as the rapid transformation of excitatory input to AP output (i.e., input–output transformation) is a hallmark of pvalb interneurons, also known as "fast in–fast out" inhibitory circuits [38,40]. Pvalb interneurons play a crucial role in complex brain functions. They rapidly inhibit other neurons and orchestrate temporally coordinated network activities that underlie cognitive processes. Their ability to rapidly and reliably inhibit other neurons

allows them to precisely control local circuits and shape brain network activity. We hypothesize that a low AP threshold coexists with slow membrane potential kinetics in human pvalb neurons to accelerate their operation in these processes.

In the present study, whole-cell electrophysiological recordings in human and mouse *ex vivo* neocortical slices revealed an association between slow somatic $V_m$ changes and a lowered AP generation threshold in pvalb/*PVALB* interneurons. The reduced AP firing threshold in human cells was accompanied by molecular specialization of the axon initial segment (AIS), a speciated axonal structure that initiates APs [41,42], which is deficient in inhibitory Kv1 potassium channels in human neurons. Simulations of fast-spiking interneurons with a human-type AIS showed that a reduced AP threshold shortens the latency of AP initiation, enabling human pvalb neurons to function as fast in–fast out circuits despite slower somatic membrane responses [43].

## Results

### Pvalb interneurons in humans exhibit a lower AP firing threshold

We measured AP firing thresholds in 91 pvalb-expressing fast-spiking interneurons [43] from human neocortical tissue resected during deep-brain surgery in 65 patients (S1 Table). The cells were studied using whole-cell clamp recordings in layer 2/3 of acute brain slices sectioned from the frontal ($n=39$ cells), temporal ($n=25$ cells), and parietal, occipital, or other neocortical regions ($n=27$ cells) in male ($n=30$) and female ($n=34$) patients aged 11–85 years. The neurons were filled with biocytin during recordings and visualized *post hoc* using streptavidin–fluorophore conjugates. All cells were immunopositive for pvalb ($n=79$; Fig 1A$_1$–1A$_3$) or expressed *PVALB* based on mRNA analysis using patch-sequencing following nucleus harvesting ($n=12$; S1 Table). For comparison, we also investigated pvalb-expressing interneurons in the mouse neocortex ($n=94$).

We applied positive $V_m$ steps (square-pulse current steps of increasing amplitude, 500 ms at 1 Hz) to cells current-clamped at −70 mV to identify the AP threshold (Fig 1B and 1C) (defined as the $V_m$ at which the AP initial rise speed reached 10 mV/ms; hereafter, referred to as the firing threshold; Fig 1A$_1$) [31,36,44]. Human pvalb neurons had a lower AP threshold [median = −38.9 mV, interquartile range (IQR) = −42.8 to −34.2, $n=91$ cells] compared with mouse neurons (median = −32.5 mV, IQR = −35.9 to −28.3, $n=94$ cells, $p<0.001$; Fig 1D$_1$ and 1D$_2$). They also exhibited a lower rheobase current (human cells: 87.34 pA, IQR = 16.8–140.0), which is the amplitude of current needed to reach the AP threshold, compared to mouse cells (247.65 pA, IQR = 53.95–418.4; $p<0.001$). Human cells showed a moderately lower AP initial rise slope (human cells: 30.9 ms$^{-1}$, IQR = 23.82–35.50; mouse cells: 33.2 ms$^{-1}$, IQR = 28.90–39.01; $p=0.0011$). This parameter determines the angle of the tangent line fitted to the early rise of the AP waveform. It measures the AP component generated in the AIS. Human cells had a slower maximum fall speed than mouse cells (human cells: −122.83 mV/ms, IQR = −165.0 to −82.7; mouse cells: −153.7 mV/ms, IQR = −213.9 to −113.7; $p=0.001$). However, the maximum rise speed of APs was similar across species (human cells: 246.94 mV/ms, IQR = 170.52–335.95; mouse cells: 255.33 mV/ms, 202.10–302.77; $p=1.00$), and no differences were observed in AP peak values ($p=1.00$). These results demonstrate that the AP repolarization phase, mediated by an active K$^+$ current, is faster in mice than in human neurons. Consequently, it is logical that human neurons also have a wider AP half-width ($p<0.01$). Compared to mouse neurons, human cells exhibited shorter latency to AP generation at rheobase ($p<0.001$), suggesting that they have specific mechanisms that facilitate AP firing and shorten the AP lag. These AP properties and interspecies comparisons are summarized in Table 1 (analyzed via PERMANOVA and Mann–Whitney $U$ test with Bonferroni correction).

We confirmed that AP thresholds in human neurons were similar between frontal (−39.0 mV, IQR = −42.2 to −30.6, $n=39$ cells) and temporal cortices (−38.5 mV, IQR = −45.1 to −36.0, $n=25$ cells; $p=0.248$, Mann–Whitney $U$ test; Fig 1E$_1$). Average AP threshold per subject showed no correlation with demographic factors. Across patients, AP firing thresholds were similar in males (−38.8 mV, IQR = −43.4 to −32.8, $n=32$ patients) and females (−40.5 mV, IQR = −45.3 to −37.3, $n=33$ patients; $p=0.405$, Mann–Whitney $U$ test; Fig 1E$_2$) and unrelated to age ($r=−0.152$, $n=63$ patients; $p=0.233$, Spearman's correlation; Fig 1E$_3$). These results show that human pvalb neurons exhibit a lower AP firing threshold in response to somatic excitation compared with mouse pvalb neurons.

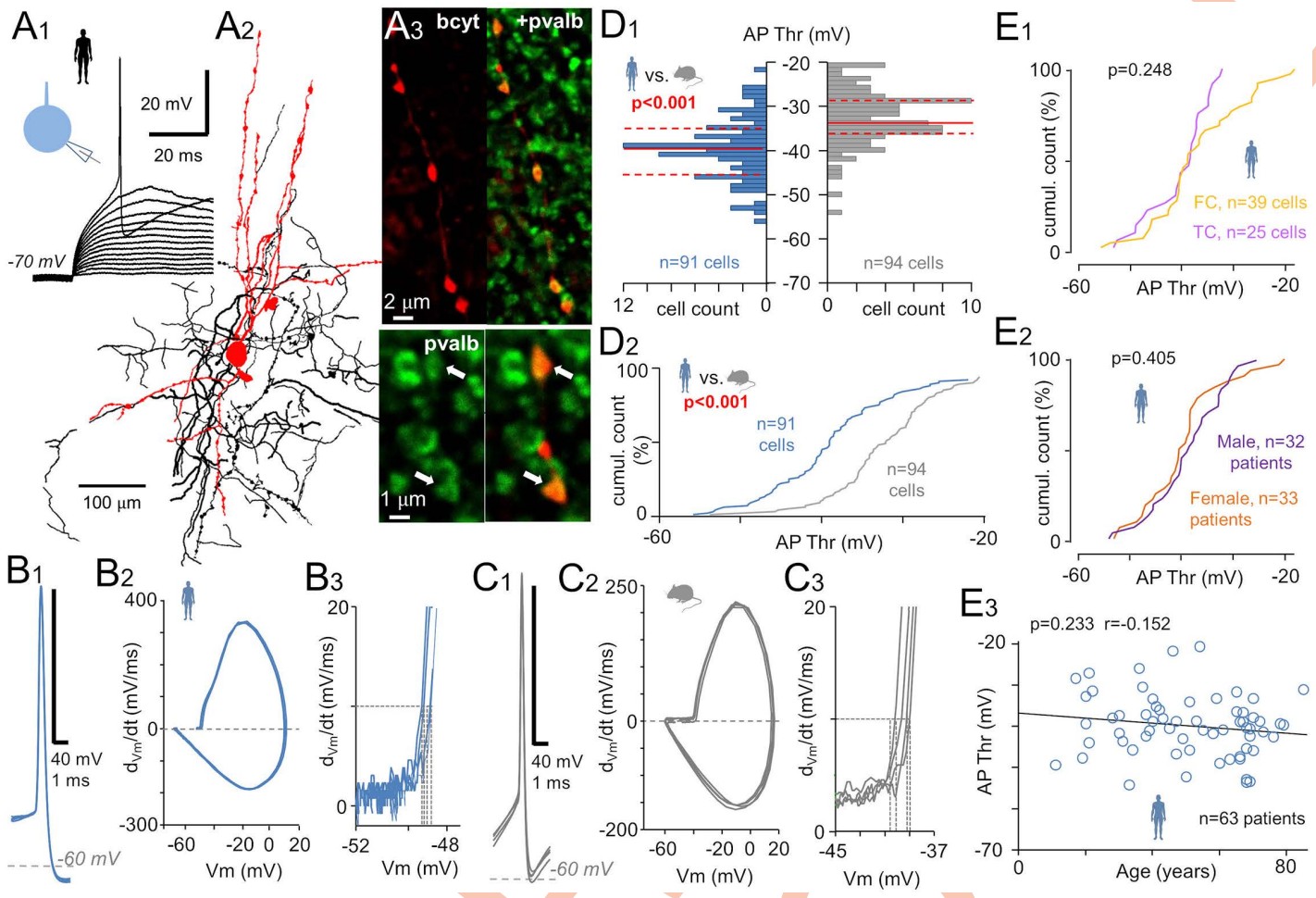

**Fig 1. Lowered AP firing threshold in fast-spiking pvalb interneurons in the human vs. mouse neocortex. (A)** AP firing threshold was examined in layer 2/3 interneurons showing a positive immunoreaction for pvalb. **(A$_1$)** Excitatory $V_m$ steps (incremental amplitude steps under whole cell current–clamp conditions at −70 mV) revealed the AP threshold. **(A$_2$)** Human neocortex pvalb neurons were visualized postrecording; the soma and dendrites (black) and axon (red) were filled with biocytin and are shown in three merged 50-µm-thick sections. **(A$_3$)** Confocal images illustrate pvalb immunoreactivity (Alexa 488) in the cell shown in (A$_2$) and a biocytin-filled axon (bcyt, Cy3). Arrows mark colocalized signals. **(B, C)** AP phase plot defining the firing threshold. **(B$_1$)** Four superimposed APs evoked as in A$_1$ in human pvalb cells. **(B$_2$)** Phase plots showing the first derivative (*dV/dt*) against $V_m$ (four superimposed curves). **(B$_3$)** Close-up of the initial AP rise; dotted vertical lines indicate threshold $V_m$ where (*dV/dt*) reaches 10 mV/ms (horizontal dotted line). **(C$_1$)** Mouse pvalb cell APs (four superimposed), with corresponding phase plots **(C$_2$)** and the firing threshold definition **(C$_3$)**. **(D)** Human pvalb neurons show a lower AP threshold compared with mouse neurons (Mann–Whitney U test). **(D$_1$)** AP threshold histograms for human (blue, 91 cells) and mouse (gray, 94 cells) neurons; bin size: 1 mV. Red lines denote median (solid) and quartiles (dotted). **(D$_2$)** Cumulative histograms of AP threshold in human (blue) and mouse (gray) pvalb cells. **(E)** In human pvalb neurons, AP threshold was not influenced by **(E$_1$)** neocortical region [*n* = 39 cells, frontal cortex (FC); *n* = 25 cells, temporal cortex (TC)], **(E$_2$)** sex (threshold calculated as the average per patient; 33 females, 32 males; Mann–Whitney *U* test), or **(E$_3$)** age (average per patient; *n* = 63; Spearman's correlation). Regression line is shown. The data underlying this Figure can be found in S1 Data.

## Low AP threshold is associated with electrically slow somata in pvalb interneurons

Compared with mouse pvalb neurons, human pvalb neurons exhibited higher electrical resistance, i.e., input resistance (Rin) in their somata, with no difference in capacitance (Cm). We measured average Rin and Cm using subthreshold $V_m$ steps of +5 to +10 mV (elicited by square-pulse currents at −70 mV; Fig 2A$_1$). Median Rin was 187.9 MΩ

**Table 1. Features of human and mouse APs from Fig 1.**

| | Human *n* = 91 | | | Mouse *n* = 94 | | | |
|---|---|---|---|---|---|---|---|
| | **Median** | **1st quartile** | **3rd quartile** | **Median** | **1st quartile** | **3rd quartile** | **P-value** |
| **AP Thr (mV)** | −38,9 | −42,8 | −34,2 | −32,5 | −35,9 | −28,3 | **<0.001** |
| **Initial rise slope (ms⁻¹)** | 30,9 | 23,8 | 35,5 | 33,2 | 28,9 | 39,0 | **0.037** |
| **AP peak $V_m$ value (mV)** | 15,6 | 9,08 | 25,0 | 15,9 | 8,2 | 23,5 | 1 |
| **AP half-width (ms)** | 0,496 | 0,407 | 0,676 | 0,362 | 0,320 | 0,416 | **<0.001** |
| **AP rheobase (pA)** | 87,4 | 47,5 | 140,0 | 247,6 | 142,2 | 418,4 | **<0.001** |
| **AP latency (ms)** | 19,6 | 12,6 | 33,7 | 50,5 | 17,9 | 79,9 | **<0.001** |
| **Spike number by step** | 3,8 | 1,8 | 11,8 | 5,0 | 2,5 | 8,0 | 1 |
| **AP AHP ampl (mV)** | −19,7 | −16,2 | −23,6 | −18,6 | −15,9 | −21,1 | 0.7794 |
| **Max rise speed (mV/ms)** | 246,9 | 170,5 | 335,9 | 255,3 | 202,1 | 320,7 | 1 |
| **Max fall speed (mV/ms)** | −122,8 | −165,0 | −82,7 | −153,7 | −213,9 | −113,7 | **0.001** |

AP Thr: action potential firing threshold; initial rise slope: maximum slope of the tangent line in the AP phase plot above the threshold (10 mV/ms); AP half-width: measured at 50% of the maximal amplitude from the AP firing threshold to the peak; AP rheobase: amplitude of depolarizing current required to trigger an AP at the AP Thr; AP latency: average time from rheobase current onset-to-AP onset (defined at AP Thr); spike number: number of APs evoked by the rheobase current step; AP AHP ampl.: amplitude of afterhyperpolarization, measured from AP Thr to AP negative peak; max rise speed: maximum AP rise speed, measured from the AP phase plot; max fall speed: maximum AP fall speed, measured from the AP phase plot). Bolded *p*-values indicate species differences (PERMANOVA and Mann–Whitney *U* test with Bonferroni correction).

(IQR = 103.8–279.8) in human cells (*n* = 60) versus 125.7 MΩ (IQR = 89.5–167.5) in mouse cells (*n* = 50; *p* < 0.001, Mann–Whitney *U* test; Fig 2A₂). Median Cm in the soma was similar across species (humans: 32.4 pF, IQR = 24.3–53.7, *n* = 60; mice: 35.1 pF, IQR = 26.3–50.6, *n* = 50; *p* = 0.728, Mann–Whitney *U* test).

The time course of $V_m$ changes (apparent tau, i.e., "tau"), measured from the ascending phase of +10 mV steps, was slower in human pvalb neurons (median "tau" of the $V_m$ step: 6.65 ms, IQR = 4.22–8.17, *n* = 60) than in mouse pvalb neurons (median: 4.17 ms, IQR = 3.27–6.12, *n* = 50; *p* < 0.001, Mann–Whitney *U* test; Fig 2B₁ and 2B₂). In mice, "tau" and Rin showed a significant positive correlation (*r* = 0.613, *n* = 50, *p* < 0.001; Fig 2C₁), whereas no such correlation was found in human pvalb neurons (*r* = 0.224, *n* = 60, *p* = 0.085) (Fig 2C₂). However, in both species, lower AP thresholds were associated with slower "tau" (human cells: *r* = −0.274, *n* = 60, *p* = 0.034; mouse cells: *r* = −0.535, *n* = 50, *p* < 0.001; Fig 2D₁ and 2D₂). Combined data from both species revealed a strong negative correlation between AP threshold and slow "tau" (*r* = −0.514, *n* = 110; *p* < 0.001, Spearman's correlation; Fig 2D₃).

Collectively, these results demonstrate that pvalb neurons with slower $V_m$ kinetics to depolarization in the soma tend to have lower AP firing thresholds. This association was particularly strong when analyzing human and mouse pvalb neurons together.

**AIS is moderately elongated in human pvalb neurons and exhibits an axonal position associated with AP threshold**

To explore whether AIS morphology contributes to the lower AP threshold in human pvalb neurons, we visualized the AIS, a primary site of AP initiation, as its length and position can affect AP threshold [45]. We used immunohistochemistry against beta-IV-spectrin, a cytoskeletal protein enriched in the AIS [42,46,47], analyzing its immunofluorescence in three-dimensional (3D) confocal microscopy image stacks, with AIS length and distance from the soma quantified using a custom script. All cells were located in layer 2/3 of the neocortex and confirmed as pvalb-positive (Fig 3A₁).

Both human and mouse pvalb neurons exhibited cell-to-cell variation in AIS length and positioning along the axon (Fig 3A₂). The median AIS start point was similar between species (humans: 4.12 μm, IQR = 2.95–5.28, *n* = 100 cells in

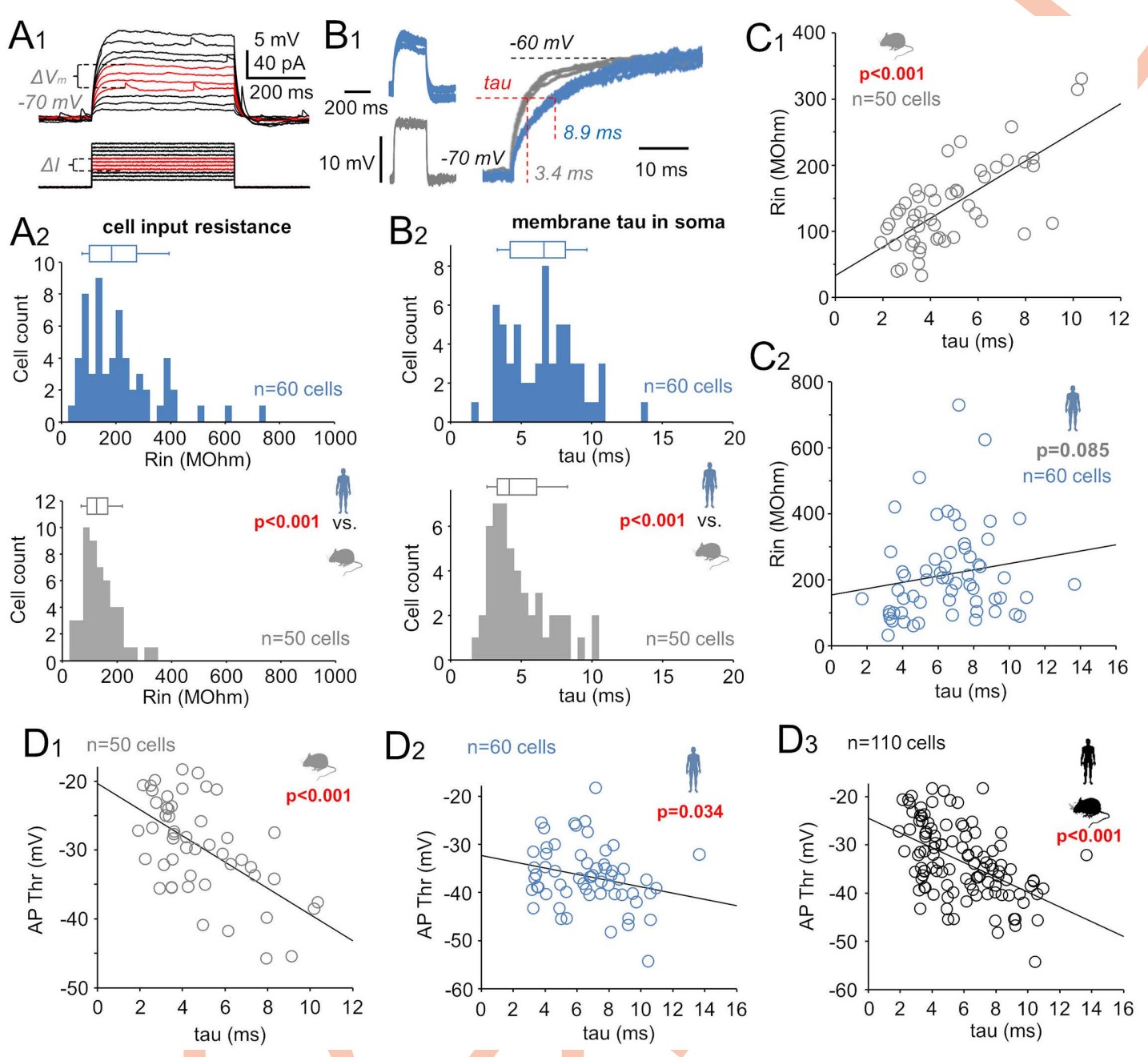

**Fig 2. Low AP firing threshold in pvalb interneurons is associated with an electrically slow soma. (A)** Human pvalb cells exhibited higher Rin compared with mouse cells at $V_m$ between −70 mV and the AP threshold. **($A_1$)** Cell Rin was measured using positive current steps inducing 5–10-mV subthreshold depolarizations. Voltage shifts ($\Delta V_m$) and currents steps ($\Delta I$) used to calculate average Rin are shown in red. **($A_2$)** Histograms and box plots show Rin in human (blue) and mouse (gray) pvalb neurons; bin size: 25 MΩ; box plots present the median, quartiles, and 5th and 95th percentiles (Mann–Whitney $U$ test). **(B)** "Tau" was longer in human than in mouse pvalb cells for $V_m$ shifts in the soma. **($B_1$)** "Tau" was measured from the ascending phase in +10-mV steps delivered at −70 mV (human: blue; mouse: gray). Left: Four superimposed +10-mV steps in human and mouse cells. Right: Ascending phase shows species differences in "tau" (human average: 8.9 ms; mouse average: 3.4 ms). **($B_2$)** Histograms and box plots illustrate "tau" differences in human (blue) and mouse (gray) pvalb neurons; bin size: 0.5 ms (Mann–Whitney $U$ test). **(C)** Soma "tau" and Rin were strongly associated in mouse **($C_1$)** but not human **($C_2$)** pvalb neurons (Spearman's correlation). **(D)** Low AP threshold was linked with slow "tau" in mouse **($D_1$)** and human **($D_2$)** pvalb neurons. **($D_3$)** Combined data from both species revealed a strong association between slower "tau" and lower AP threshold (Spearman's correlation). The data underlying this Figure can be found in S1 Data.

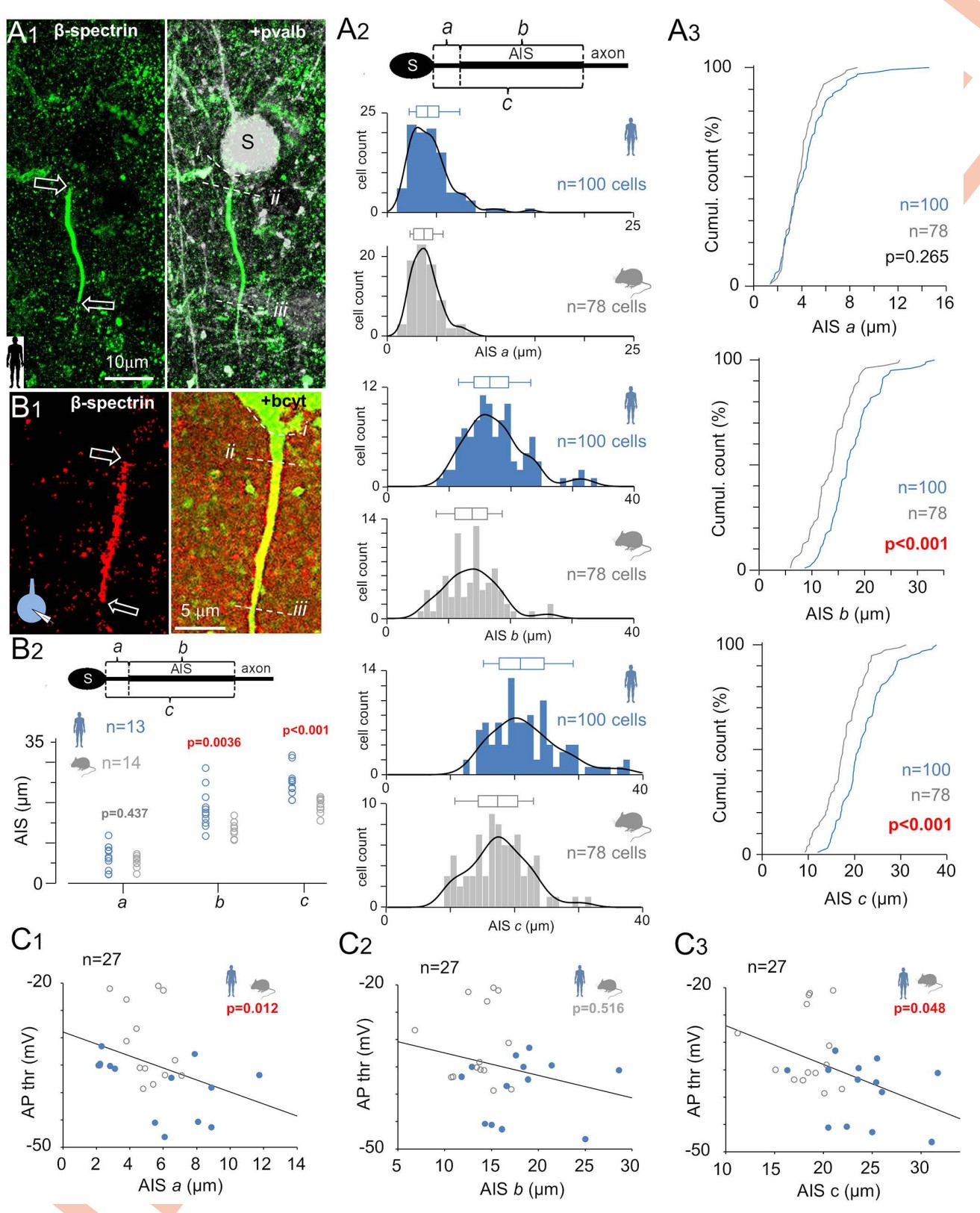

**Fig 3. AIS is elongated in human pvalb neurons and axonally located according to AP threshold. (A)** AIS in pvalb neurons was moderately longer in humans than in mice. AIS analysis was performed in triple-immunostained neurons from the human and mouse neocortex. **(A$_1$)** AIS was measured in 3D fluorescence confocal images of layer 2/3 pvalb-positive neurons. Left: AIS visualized via beta-IV-spectrin immunofluorescence. Arrows indicate AIS initiation and termination sites (Alexa 488) in a collapsed z-stack image of a human pvalb neuron. Right: Merged beta-IV-spectrin and pvalb (Cy5) immunoreactions. Horizontal dotted curves mark (i) the axon from the soma (s) and the proximal (ii) and (iii) distal AIS ends. **(A$_2$)** Histograms and box plots summarize AIS parameters from the upper schematic: a = distance from the soma to the AIS initiation site, b = AIS length, and c = distance from the soma to the AIS end. Top: AIS initiation distance from the soma did not differ between species. Middle: AIS was longer in human neurons than in mouse neurons. Bottom: AIS termination site was more distal in human neurons than in mouse neurons. Histogram bin: 1 μm, with kernel density; box plots show the median, quartiles, and 5th and 95th percentiles. **(A$_3$)** Cumulative AIS data show p-values (Student t test). **(B, C)** AIS measurements in electrophysiologically investigated pvalb neurons revealed that a low AP threshold is associated with a more distal AIS termination site. **(B$_1$)** AIS visualized via beta-IV-spectrin immunopositivity (Cy3) in human and mouse cells filled with biocytin (bcyt) and labeled with streptavidin–Alexa 488 conjugates. Immunofluorescence images show AIS features defined in A1. **(B$_2$)** Electrophysiologically studied human pvalb neurons displayed a longer AIS (b) with a more distal termination site (c) compared with mouse neurons, whereas the AIS start distance (a) from the soma did not differ between species (Mann–Whitney U test). Combined AP threshold data from human (blue) and mouse (gray) pvalb neurons revealed associations with AIS initiation site **(C$_1$)** and termination site **(C$_3$)** distance from the soma, but no association with AIS length **(C$_2$)** (Spearman's correlation). The data underlying this Figure can be found in S1 Data.

17 patients; mice: 3.77 μm, IQR = 2.69–4.68, n = 78 cells in 15 mice; p = 0.265), but the AIS was longer in human cells (median: 16.68 μm, IQR = 14.08–19.78) than in mouse cells (median: 13.99 μm, IQR = 11.15–16.55; p < 0.001). Consequently, the AIS termination site was further from the soma in human neurons (21.01 μm, IQR = 17.60–24.66, n = 100 cells from 17 samples) than in mouse neurons (17.47 μm, IQR = 14.41–20.66 μm, n = 78 cells from 15 mice; p < 0.001, Mann–Whitney U test; Fig 3A$_2$ and 3A$_3$).

We also investigated whether these morphological differences relate to the lower AP threshold observed in humans. In electrophysiologically studied and biocytin-filled neurons visualized via streptavidin–fluorophore conjugates, with confirmed immunopositivity for pvalb, we observed the AIS based on beta-IV-spectrin positivity and measured its length and distance from the soma along the axon, comparing these metrics against the AP firing threshold. Of the 28 human and 22 mouse cells recorded in the whole-cell clamp, intact somata and full-length AIS were successfully recovered in 13 human and 14 mouse pvalb immunopositive neurons. The total number of cells includes all cells that were electrophysiologically recorded and subsequently visualized via biocytin staining. Cells that failed include pvalb cells with ruptured or damaged somata, failure to confirm continuation of the axon after the beta-IV-spectrin-labeled AIS, and failure to clearly measure AIS parameters due to an unsuccessful immunoreaction in human tissue. The double immunofluorescence reaction had to be successful for the pvalb and beta-IV-spectrin antibodies, and the axon had to initiate from the soma with a fully visualized AIS. A complete AIS was confirmed if the biocytin-filled axon continued beyond the beta-IV-spectrin-labeled AIS (Fig 3B$_1$). The distance from the AIS's initial point to the soma in human pvalb neurons was 6.10 μm (IQR = 2.55–8.50, n = 13) compared with 4.85 μm (IQR = 3.80–6.03, n = 14) in mouse neurons (p = 0.437; Fig 3B$_2$). AIS length was greater in human cells (17.60 μm, IQR = 14.65–20.20 μm) than in mouse cells (14.0 μm, IQR = 12.10–15.33; p = 0.004), and the AIS terminal site was farther from the soma in human neurons (23.60 μm, IQR = 20.85–25.75) than in mouse neurons (18.55 μm, IQR = 17.68–20.38; p < 0.001, Mann–Whitney U test). All AIS properties are presented in Fig 3B$_2$. None of the AIS parameters were correlated with the AP threshold in either species. The correlation between AIS proximal distance and AP threshold was 0.536 (p = 0.054) in human cells (n = 13) and −0.297 (p = 0.93) in mouse cells (n = 14). AIS length was not significantly correlated with AP threshold in humans (r = 0.203, p = 0.493) or mice (r = 0.079, p = 0.773), nor was distance from the AIS termination site to the soma correlated with AP threshold in either species (humans: r = −0.239, p = 0.414; mice: r = 0.024, p = 0.988, Spearman's correlation). However, when pooling data across species, the AP threshold was significantly correlated with AIS start distance (r = −0.478, p = 0.012) and AIS termination distance (r = −0.383, p = 0.0485) but not AIS length (r = −0.1340, p = 0.516, Spearman's correlation; Fig 3C$_1$–3C$_3$).

Because the number of cells studied electrophysiologically and investigated anatomically is relatively small, and because the range of measured AP threshold (AP Thr) and AIS structure values is large, it is possible that our dataset

is not large enough to reveal hidden associations between AIS geometry and AP threshold within each species. Nevertheless, these findings suggest that the difference in AP threshold between human and mouse pvalb neurons cannot be explained by AIS length or axonal location. However, they suggest that the position of the AIS may partly explain the large variability observed in the AP threshold between individual neurons. Specifically, a more distal AIS placement is associated with a lower AP threshold.

## Axon diameter does not account for interspecies differences in AP threshold

We examined axon diameter in biocytin-filled pvalb interneurons to assess its potential correlation with the AP firing threshold, particularly across species, as axon caliber can influence axon excitability. Thicker axons have lower longitudinal resistance (Ra) in the AIS. This allows for faster passive electrical conduction between the soma and the AIS. However, the lowered resistance in the axon simultaneously increases the AIS's AP rheobase current. However, larger axons also have more ion channels, including voltage-gated sodium channels that initiate the AP, suggesting a complex relationship between axon diameter and AP Thr [48]. Axonal diameters were measured using high-magnification fluorescence confocal microscopy images of biocytin-filled neurons at five standardized positions along the axon: two pre-AIS sites and three sites within the AIS (Fig 4A$_1$–4A$_3$). Reported data represent the mean values per cell for the pre-AIS and within-AIS portions of the axon. No significant differences were observed in axon diameter between human ($n = 12$) and mouse ($n = 14$) pvalb neurons in either in the pre-AIS [1.07 μm (IQR = 0.82–1.19) versus 0.88 μm (IQR = 0.71–1.14); $p = 0.939$] or within-AIS portion [0.83 μm (IQR = 0.70–0.96) versus 0.69 μm (IQR = 0.64–0.74); $p = 0.260$; one-way ANOVA on ranks; Fig 4B]. Additionally, in both AIS portions, axon diameter was not correlated with AP threshold in humans (pre-AIS: $r = 0.245$, $p = 0.429$; within-AIS: $r = 0.392$, $p = 0.197$; $n = 12$) or mice (pre-AIS: $r = 0.091$, $p = 0.766$; within-AIS: $r = 0.056$, $p = 0.852$; $n = 12$). Pooled data across species also showed no association (pre-AIS: $r = 0.137$, $p = 0.518$; within-AIS: $r = -0.036$, $p = 0.866$, Spearman's correlation; Fig 4C$_1$ and 4C$_2$). Collectively, these results indicate that axon diameter does not contribute to the observed interspecies differences in AP threshold in pvalb neurons.

## Nav1.6 channels are similarly distributed in the AIS of human and mouse pvalb neurons

We examined immunoreactivity for the low-voltage-gated sodium channel Nav1.6 in the soma and axon of pvalb neurons (Fig 5A–5H), as Nav1.6 density and spatial distribution in the AIS can regulate AP threshold [49,50]. Nav1.6 may concentrate at the AIS's distal end to lower the AP firing threshold [48,49,51,52], or it may be evenly distributed. In layer 2/3 of the human and mouse neocortex, we assessed Nav1.6 immunofluorescence (Fig 5A$_1$ and 5A$_2$) intensity using confocal microscopy images of the soma (Fig 5B$_1$ and 5B$_2$) and 3D image stacks along the axon (Fig 5C$_1$ and 5C$_2$) to evaluate species-specific differences in Nav1.6 channel distribution. In individual human cells ($n = 21$; Fig 5D$_1$) and mouse cells ($n = 26$; Fig 5D$_2$), 24 radial lines were drawn from within the soma to the extracellular space to measure Nav1.6 (Cy3) and pvalb (Cy5) immunofluorescence. Because pvalb is intracellular, its onset along each line defined the extracellular membrane site (Fig 5D$_1$ and 5D$_2$). We also measured immunofluorescence intensity for Nav1.6, pvalb, and beta-IV-spectrin in the same cells along the axon, annotated in 3D image stacks, from the soma to beyond the AIS termination site (Fig 5E$_1$ and 5E$_2$).

Reference-corrected Nav1.6 immunofluorescence ($F-F_{ex.c}$, defined as the fluorescence intensity with the extracellular average subtracted from other radial or axon pixels) was similar across species in the soma membrane (humans: 7.06, IQR 4.72–15.12, $n = 21$; mice: 8.52, IQR = 3.30–12.50, $n = 26$; $p = 1.00$) and the cytoplasm (humans: 23.66, IQR = 16.37–31.64, $n = 21$; mice: 12.39, IQR = 5.88–20.25, $n = 26$; $p = 0.067$).

Nav1.6 immunofluorescence was similarly weak in human and mouse pre-AIS axons (humans: 6.55, IQR = 4.03–20.69, $n = 21$; mice: 6.25, IQR = −0.87–16.24, $n = 26$; $p = 1.00$) and post-AIS regions (humans: 0.95, IQR = −3.41 to 7.56; mice: 4.59, IQR = −2.49 to 13.13; $p = 1.00$). Nav1.6 intensity was strong in the AIS but not significantly different between humans (73.18, IQR = 49.32–94.05) and mice (48.40, IQR = 28.84–67.81; $p = 0.063$, PERMANOVA and Mann–Whitney $U$ test with Bonferroni correction; Fig 5E$_1$ and 5E$_2$). In both species, Nav1.6 immunofluorescence was uniformly distributed within the

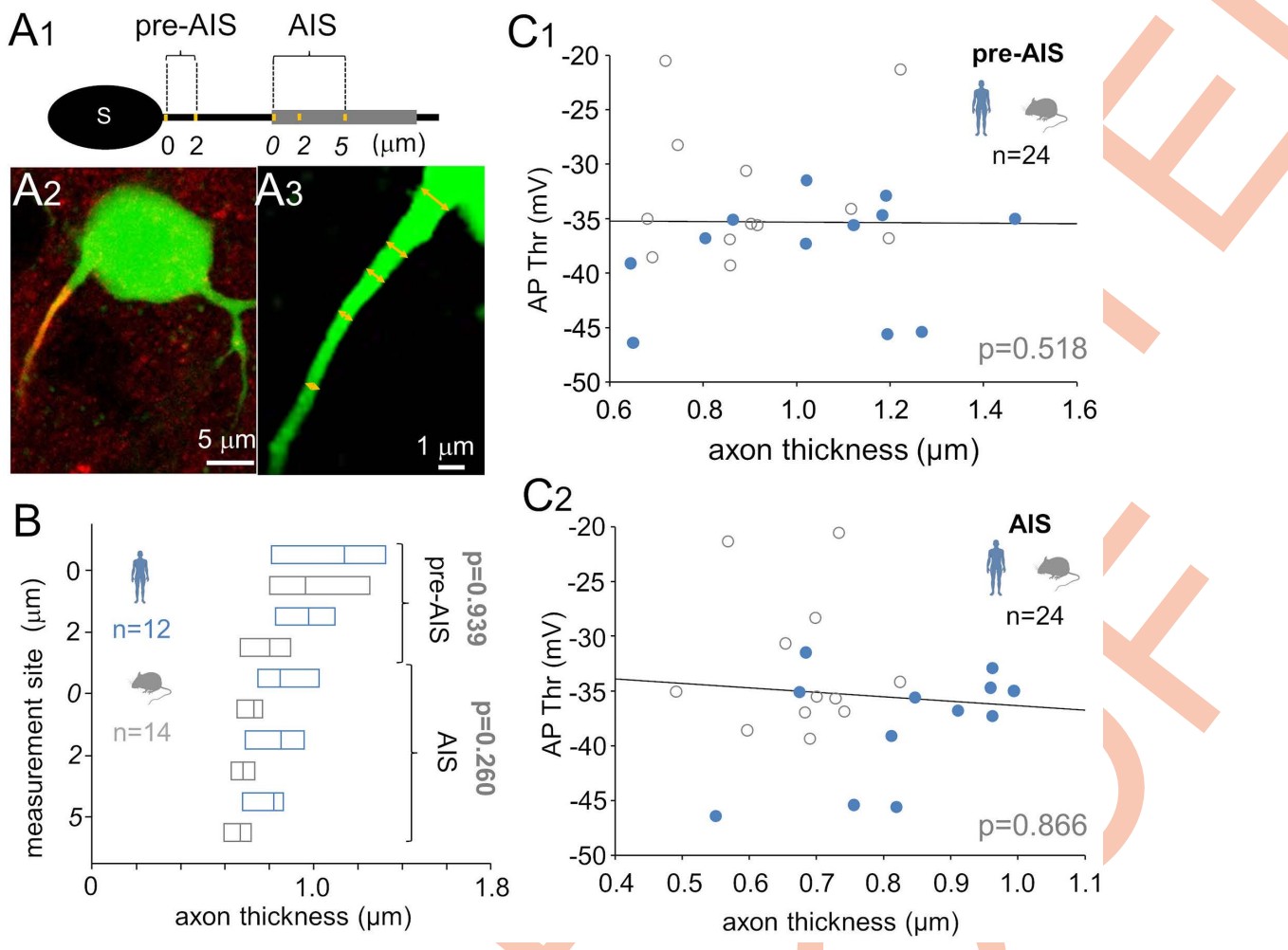

**Fig 4. Axon diameter fails to explain interspecies differences in pvalb interneuron AP firing threshold. (A)** Axon cross-diameter was measured in 3D confocal images of biocytin-filled pvalb interneurons. **(A₁)** Schematic showing axon maximum width at five points. "Pre-AIS" measurements were taken at the axon's start (0 μm from the soma) and 2 μm from the soma. "AIS" measurements were taken at the AIS start (shown as *0*) and at *2* and *5* μm from the AIS initiation site. **(A₂)** Collapsed confocal image showing biocytin-filled human pvalb neurons (Alexa 488), with the AIS visualized via beta-IV-spectrin immunoreactivity (Cy3). **(A₃)** Enlarged image of the axon highlighting five measurement sites. **(B)** Axon thickness was similar between human (blue) and mouse (gray) pvalb neurons. Horizontal bars indicate medians and quartiles. **(C)** Axon diameter was not correlated with AP threshold in either the pre-AIS region **(C₁)** or the AIS **(C₂)**. Data from both species were pooled (one-way ANOVA on ranks with Dunn's pairwise *post hoc* test). The data underlying this Figure can be found in S1 Data.

AIS, with no difference between the first and second halves (humans: $p = 0.182$; mice: $p = 0.833$, Mann–Whitney $U$ test; Fig 5E₁ and 5E₂). Nav1.6 $F$–$F_{ex.c}$ values are shown in Fig 5F for humans and Fig 5G for mice, with is no species differences observed (Fig 5H).

## Human pvalb neurons exhibit low Kv1 channel levels in the AIS relative to mouse neurons

We investigated the immunofluorescence of Kv1.1 and Kv1.2 potassium channels, key inhibitory mechanisms modulating AP threshold in the AIS [53,54]. Kv1 channels are clustered in the AIS and soma of rodent pvalb neurons [55], where they increase the AP firing threshold [53,56] (hereafter, Kv1 channels are referred to as inhibitory channels). First, we analyzed Kv1.1 immunofluorescence in the soma and AIS of human (Fig 6A₁) and mouse (Fig 6A₂) pvalb neurons using triple

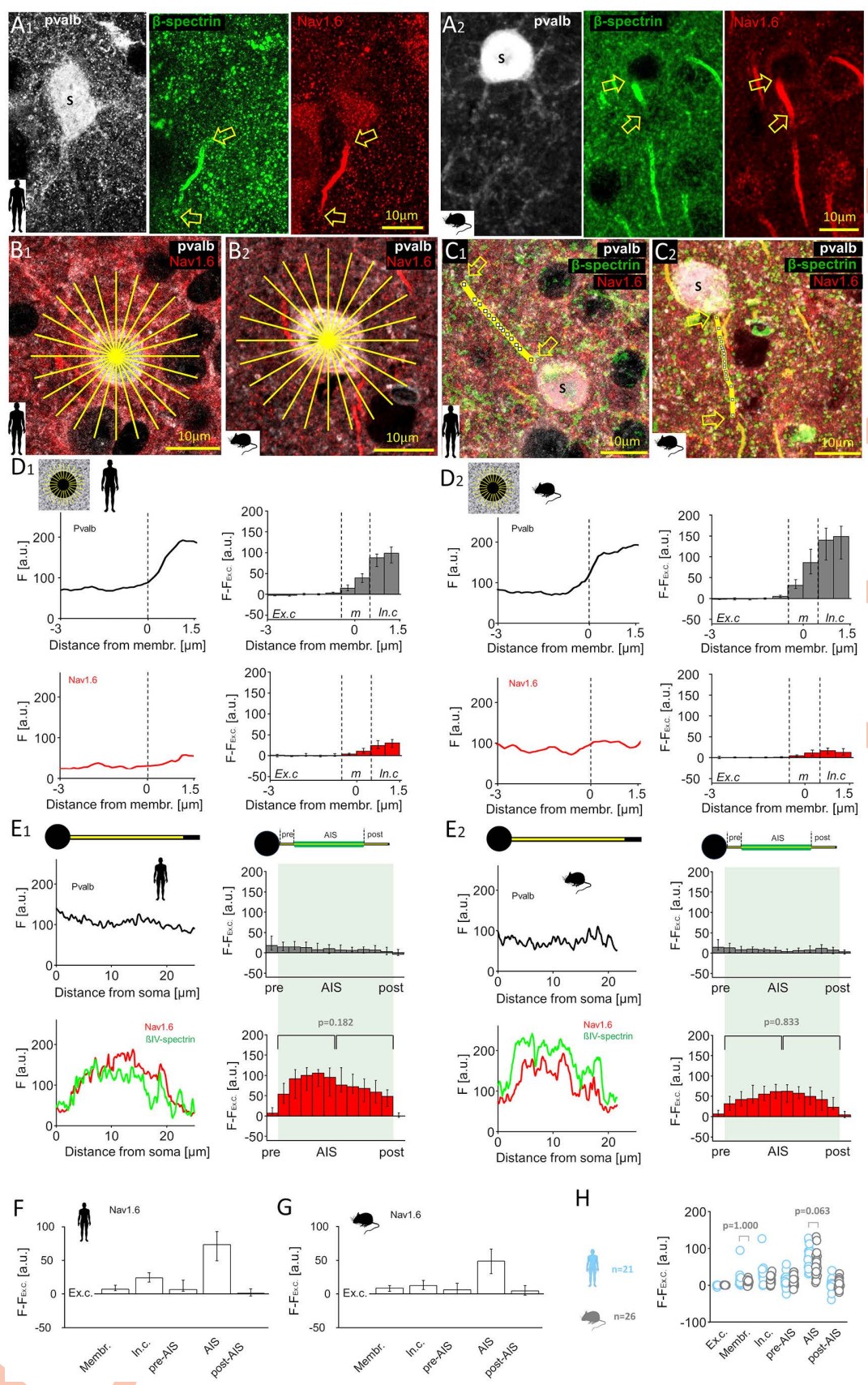

**Fig 5. AIS displays robust and uniform immunofluorescence for Nav1.6 channels in human and mouse pvalb neurons. (A)** Triple immunofluorescence for pvalb (Cy5), beta-IV-spectrin (Alexa 488), and Nav1.6 (Cy3) revealed sodium channel localization in the AIS of human $(A_1)$ and mouse $(A_2)$ pvalb neurons. Arrows mark AIS initiation and termination sites; s indicates the soma. Images are collapsed z-stacks from layer 2/3 of the neocortex. **(B)** Nav1.6 immunofluorescence in the somatic cytoplasm and membrane was analyzed alongside pvalb in human $(B_1)$ and mouse $(B_2)$ cells. Immunofluorescence intensity was measured using a 24-line radial pattern (yellow lines) extending from the extracellular space to the soma center. **(C)** Nav1.6 intensity was measured along the axon from the soma (s) past the AIS termination site in 3D confocal images. Measurement lines in human $(C_1)$ and mouse $(C_2)$ pvalb neurons overlie beta-IV-spectrin–defined AIS (masked in the images by the lines). Images are collapsed from z-stacks. **(D)** Immunofluorescence intensity [ordinate, arbitrary unit (a.u.)] was measured for pvalb (top, black line) and Nav1.6 (bottom, red line) along radial lines spanning the extracellular space (negative abscissa values) to the soma (positive values). Vertical dashed line denotes the membrane location (0-point), defined as ±0.5 µm from the first pvalb signal. $(D_1)$ Left: Mean fluorescence intensity ($F$, a.u.) across 24 radial lines in one human cell, spanning −3 to +1.5 µm. Schematic illustrates measurement geometry. Right: Medians and quartiles of intensity values minus the extracellular reference ($F-F_{ex.c}$) in 21 human cells. $(D_2)$ Identical measurements as in $D1$ for one mouse cell (left) and 26 mouse cells (medians and quartiles; right). **(E)** Fluorescence intensity along 3D pixel lines for pvalb (black), beta-IV-spectrin (green), and Nav1.6 (red). $(E_1)$ Left: Measurement in one human neuron (a.u.); 0 on abscissa = axon origin. Upper schematic illustrates the measurement line along the axon. Right: Medians and quartiles of referenced intensity values ($F-F_{ex.c}$) in 21 human cells. Upper schematic indicates the defined AIS (determined via beta-IV-spectrin immunofluorescence) and pre-AIS and post-AIS axonal regions. AIS length (green background) was divided into 10 bins per cell. Statistical analysis showed no difference in Nav1.6 intensity between proximal and distal AIS halves (Mann–Whitney $U$ test). $(E_2)$ Same measurements shown in $E1$ for one mouse cell (left) and 26 mouse cells (right; medians and quartiles). **(F, G)** Summary of line analyses in human and mouse pvalb neurons; medians and quartiles of $F-F_{ex.c}$ across five cellular domains showed strong Nav1.6 immunoreactivity in the AIS. **(H)** $F-F_{ex.c}$ values across domains did not differ between human (blue) and mouse (gray) pvalb cells (PERMANOVA with Bonferroni *post hoc* test). The data underlying this Figure can be found in S2 Data.

immunolabeling. Representative cells and analysis are shown in Fig 6B and 6C, including those in the soma (Fig 6B$_1$ and 6B$_2$) and along the axon (Fig 6C$_1$ and 6C$_2$). $F-F_{ex.c}$ analysis showed absent Kv1.1 signal at the human soma membrane (−1.23, IQR = −4.77 to 5.25, $n$ = 31) as well as its presence in mice (5.42, IQR = 0.62–10.58, $n$ = 31; $p$ = 0.041). Kv1.1 signal in the cytoplasm was weak in both species and did not differ significantly ($p$ = 0.817; Fig 6D$_1$ and 6D$_2$).

Axonal analysis showed robust Kv1.1 signal in the AIS of mouse cells (47.56, IQR = 13.98–74.95, $n$ = 31), consistent with previous reports [53]. In contrast, human AIS Kv1.1 signal was low or absent (8.54, IQR 5.00–24.43), despite strong signal in nearby pvalb-negative neurons (Fig 6A$_1$). AIS $F-F_{ex.c}$ differed between species ($p$ = 0.001; Fig 6E$_1$ and 6D$_2$). Pre-AIS (humans: −1.06, IQR = −6.71 to 8.77; mice: 9.56, IQR = −5.49 to 20.32; $p$ = 0.287) and post-AIS (humans: 2.55, IQR = −3.26 to 10.88; mice: 4.58, IQR = −17.97 to 29.20; $p$ = 1.00) regions showed no significant differences (PERMANOVA and Mann–Whitney $U$ test with Bonferroni correction.). Kv1.1 intensity showed no difference between AIS halves (humans: $p$ = 0.318; mice: $p$ = 0.195, Mann–Whitney $U$ test; Fig 6E$_1$ and 6E$_2$). Kv1.1 $F-F_{ex.c}$ intensity values for each domain are summarized in Fig 6F and 6G, and species comparisons are shown in Fig 6H.

We also analyzed Kv1.2 immunofluorescence (Fig 7A$_1$ and 7A$_2$), comparing $F-F_{ex.c}$ in the soma (Fig 7B$_1$ and 7B$_2$) and AIS (Fig 7C$_1$ and 7C$_2$) of human and mouse pvalb neurons [57]. A low but moderately higher Kv1.2 signal was observed at the human soma membrane (5.74, IQR = 2.08–10.42, $n$ = 60) compared with the mouse membrane (3.42, IQR = −2.66 to 5.30, $n$ = 33; $p$ = 0.013; Fig 7D$_1$ and 7D$_2$). Cytoplasmic signal was also higher in humans (4.47, IQR = −3.73 to 16.39) than in mice (−6.59, IQR = −12.07 to 4.16; $p$ = 0.002).

In the AIS, no difference in Kv1.2 $F-F_{ex.c}$ was detected between human (2.95, IQR = −3.23 to 9.05, n = 60) and mouse (17.04, IQR = −5.30 to 31.76, $n$ = 30) cells ($p$ = 0.272; Fig 7E$_1$ and 7E$_2$), nor in pre-AIS ($p$ = 0.478) or post-AIS ($p$ = 0.187) sites (PERMANOVA and Mann–Whitney $U$ test with Bonferroni correction). Kv1.2 immunofluorescence intensity showed no difference between AIS halves (humans: $p$ = 0.397; mice: $p$ = 0.984, Mann–Whitney $U$ test; Fig 7E$_1$ and 7E$_2$). Kv1.2 data are summarized in Fig 7F, 7G, and 7H.

## Pvalb neurons in humans lack *KCNA1* mRNA encoding Kv1.1 but possess *KCNA2* mRNA encoding Kv1.2

We used patch-sequencing to measure the mRNA levels of low-threshold ion channels that regulate the AP firing threshold, namely Kv1.1 and Kv1.2 [44,51,53–55,58], in electrophysiologically studied fast-spiking pvalb neurons in human and mouse neocortex layer 2/3. Whole-cell recordings were followed by nucleus extraction and transcriptomic analysis [59,60].

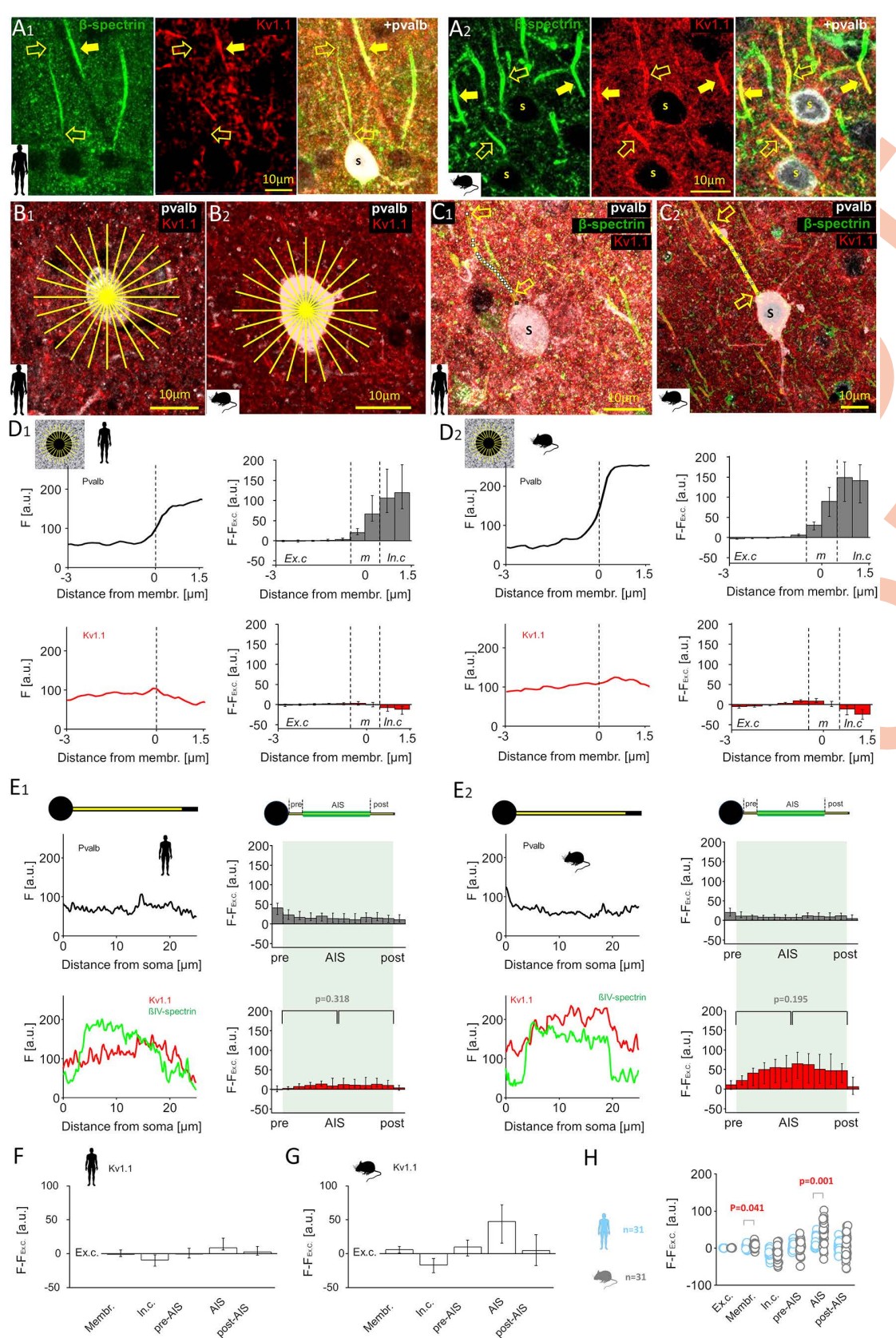

**Fig 6. AIS is deficient in Kv1.1 potassium channels in humans compared with mice. (A)** Immunofluorescence staining revealed a species-specific difference in Kv1.1 expression in the AIS of pvalb neurons. **($A_1$)** Triple labeling for beta-IV-spectrin (Alexa 488), Kv1.1 (Cy3), and pvalb (Cy5) in the human neocortex layer 2/3. Open arrows denote the AIS visualized using beta-IV-spectrin in a pvalb+ neuron; solid arrows mark the AIS of a pvalb− neuron; s denotes the pvalb+ soma. The pvalb+ neuron lacked a Kv1.1 signal, whereas the pvalb− AIS exhibited robust Kv1.1 labeling. (A2) Same triple labeling shown in *A1* in the mouse cortex; both pvalb+ (open arrows) and pvalb− (solid arrows) neurons showed robust AIS Kv1.1 expression. **(B)** Kv1.1 fluorescence intensity in somatic membranes analyzed with pvalb labeling in human **($B_1$)** and mouse **($B_2$)** neurons. Yellow radial lines indicate the 24-line measurement pattern. **(C)** Axonal Kv1.1 intensity was measured from the soma (s) in 3D confocal images in human **($C_1$)** and mouse **($C_2$)** neurons. Measurement lines are shown above the beta-IV-spectrin-labeled AIS in collapsed z-stacks. **(D)** Fluorescence intensity for pvalb (top, black line) and Kv1.1 (bottom, red line) along radial lines. Vertical dashed line marks pvalb signal onset (0-point). **($D_1$)** Left: Mean intensity (*F*, a.u.) across 24 radial lines in one human cell, spanning −3 to +1.5 μm. Schematic shows radial lines. Right: Medians and quartiles of $F - F_{ex.c}$ in 31 human cells for pvalb (gray) and Kv1.1. (red). **($D_2$)** Same measurements as in $D_1$ for one mouse cell (left) and 31 mouse cells (right; medians and quartiles). **(E)** Line analysis of axons for pvalb (black), beta-IV-spectrin (green), and Kv1.1 (red). **($E_1$)** Left: Intensity profile of one human cell, as illustrated in upper schematic (0 = the axon origin). Right: Medians and quartiles of $F - F_{ex.}$ intensity values in 31 human cells. Upper schematic shows AIS defined by beta-IV-spectrin labeling. AIS length (green background) was divided into 10 bins per cell. Statistical analysis revealed that Kv1.1 intensity does not differ between the proximal and distal AIS halves (Mann–Whitney *U* test). (E2) Identical measurements described in *E1* for one mouse neuron (left) and 31 mouse neurons (right; medians and quartiles). Statistical analysis indicated that Kv1.1. is evenly distributed along the AIS (Mann-Whitney U test). **(F, G)** Summary of line analyses in human and mouse pvalb neurons; medians and quartiles of $F - F_{ex.c}$ for Kv1.1 across cellular domains. **(H)** $F - F_{ex.c}$ values in human (blue) and mouse (gray) pvalb cells across cellular domains: mouse cells showed stronger Kv1.1 signals in the somatic membrane and AIS compared with human cells (PERMANOVA test with Bonferroni *post hoc* test). The data underlying this Figure can be found in S2 Data.

We detected mRNA for 6,784 genes (IQR = 4,872–7,399) in humans (*n* = 21) and 3,617 genes (IQR = 3,173–4,305) in mice (*n* = 16; Fig 8A). Transcriptomes were uploaded to the Allen Institute neuron type identification server, which employs an algorithm categorizing neocortex layer 2/3 neuron types by gene expression (https://knowledge.brain-map.org/map-mycells/process/), enabling identification of pvalb neurons, even in cells with undetectable *PVALB* mRNA expression. Using this method, we identified 21 and 16 human and mouse pvalb neurons, respectively. These cells were analyzed for expression of *KCNA1*, *KCNA2*, *KCNC1*, *KCNC2*, *KCND1*, *KCNQ1*, *KCNQ2*, *KCNQ3*, *KCNQ4*, *KCNQ5*, *KCNK2*, *KCNK4*, and *KCK10,* i.e., genes encoding major potassium channels in axons that regulate AP threshold [54,55]. Additionally, we analyzed *SCN8A* expression and that of the housekeeping genes *ACTB*, *GAPDH*, and *SLC8A1* (Fig 8A).

All 21 human pvalb neurons lacked detectable *KCNA1* mRNA expression, whereas 7 of 16 mouse neurons showed robust expression. The mRNA level heatmap in Fig 8A shows the gene expression (transcripts per million [(TPM]) of individual pvalb neurons in humans and mice. Comparing *KCNA1* mRNA expression between human (*n* = 21) and mouse cells (*n* = 16) revealed a significant difference (*p* = 0.0016, one-way MANOVA with a *post hoc* Bonferroni test). Furthermore, categorical comparisons (positive or negative mRNA expression) confirmed that *KCNA1* expression differed significantly between species (*p* < 0.001, chi-squared test following Benjamini–Hochberg adjustment at the 5% significance level; Fig 8A). Both species displayed *KCNA2* mRNA expression without any species-specific difference (*p* = 0.970). We also tested whether pooled mRNA levels (TPM) of *KCNA1* and *KCNA2* were associated with AP threshold in patch-sequenced neurons. No association was observed for either gene in humans (*r* = 0.041, *n* = 20, *p* = 0.861) or mice (*r* = 0.236, *n* = 15, *p* = 0.388). When neurons from both species were pooled, the results remained nonsignificant (*r* = 0.175, *n* = 35; *p* = 0.312, Spearman's correlation test).

Both species exhibited similarly strong *KCNC1* and *KCNC2* mRNA expression, which respectively encode the Kv.3.1 and Kv3.2 channels characteristics of pvalb interneurons [43,61]. They also showed equally low or undetectable expression of *KCND1* encoding Kv4.1 potassium channels [62]. No species differences were observed in *KCNK2*, *KCNK4*, or *KCNK10* mRNA levels, and *KCNQ1*, *KCNQ2*, and *KCNQ4* mRNAs were similarly expressed between species. However, human neurons displayed significantly higher mRNA levels of *KCNQ3* (*p* < 0.001) and *KCNQ5* (*p* < 0.001). Both human and mouse pvalb cells strongly expressed *SCN8A* mRNA, encoding Nav1.6 sodium channels. Additionally, all cells expressed high levels of *ACTB* and *GAPDH*. Notably, mRNA expression of *SLC6A1*, which encodes GABA transporter 1, was significantly lower in human neurons than in mouse neurons (*p* = 0.002). *PVALB* mRNA expression was undetectable in 10

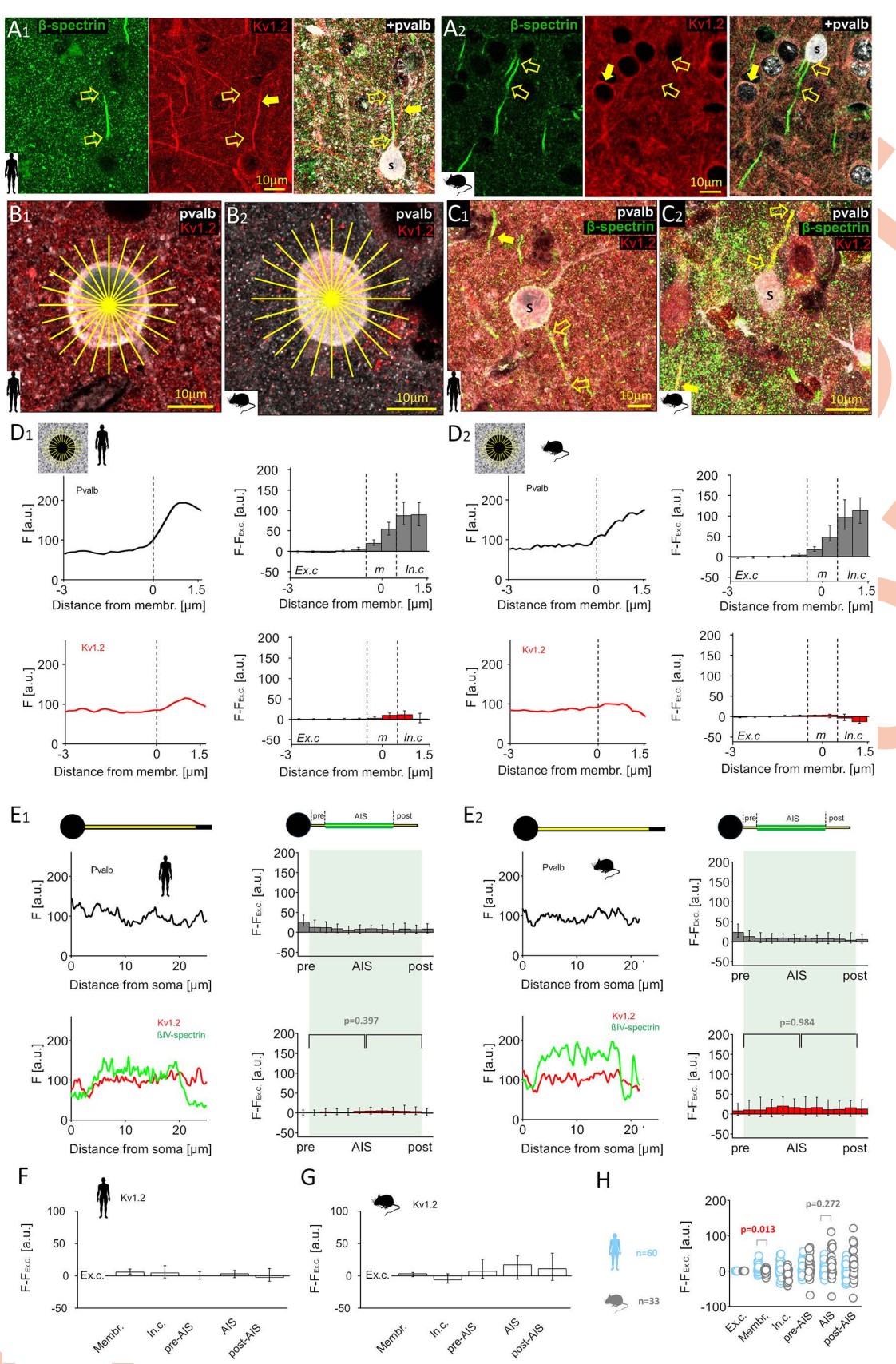

**Fig 7. Kv1.2 potassium channel expression is low in the AIS of human and mouse pvalb neurons. (A)** Triple immunofluorescence for beta-IV-spectrin (Alexa 488), Kv1.2 (Cy3), and pvalb (Cy5) revealed weak Kv1.2 immunoreactivity in the AIS of human **(A₁)** and mouse **(A₂)** pvalb neurons. Open arrows indicate AIS identified via beta-IV-spectrin in a pvalb+ neurons, which show negligible Kv1.2 signal; solid arrow denotes a Kv1.2-positive dendrite and somatic membrane of another neuron; s marks the pvalb+ soma. **(B)** Radial intensity measurements of Kv1.2 and pvalb in human **(B₁)** and mouse **(B₂)** somata using 24-line radial patterns (yellow lines). **(C)** Line measurements of Kv1.2 intensity along the axon from the soma (s) in human **(C₁)** and mouse **(C₂)** pvalb neurons. Images are collapsed z-stacks; line measurement is shown over the axon. **(D)** Immunofluorescence intensity profiles for pvalb (top, black line) and Kv1.2 (bottom, red line) along the radial lines in the soma. **(D₁)** Left: Mean fluorescence intensity ($F$, a.u.) in 24 pooled lines from one human cell, spanning −3 to +1.5 μm from the 0-point. Right: Medians and quartiles of $F - F_{ex.c}$ values for pvalb (gray) and Kv1.2. (red) in 60 human cells. **(D₂)** Identical measurements as in *D1* for one mouse cell (left) and 33 mouse cells (right; medians and quartiles). **(E)** Axonal line analysis for pvalb (black), beta-IV-spectrin (green), and Kv1.2 (red). **(E₁)** Left: Line intensity values along the axon in one human cell (illustrated in the upper schematic), with 0 denoting the axon origin. Right: Median and quartiles of $F - F_{ex.}$ values across 60 human cells. AIS length (green background) was divided into 10 bins. Statistical analysis showed similarly negligible Kv1.2 intensity in the proximal and distal AIS halves (Mann–Whitney U test). **(E₂)** Identical measurement as in *E₁* for one mouse neuron (left) and 33 mouse neurons (right; medians and quartiles), showing uniformly low Kv1.2 signal across the AIS (Mann–Whitney *U* test). **(F, G)** Summary of line analyses in human and mouse pvalb cells; medians and quartiles of $F - F_{ex.c}$ values across cellular domains. **(H)** $F - F_{ex.c}$ values in human (blue) and mouse (gray) pvalb cells: human cells showed stronger Kv1.2 signals in the somatic membrane compared with mouse cells (PERMANOVA test with Bonferroni *post hoc* test). The data underlying this Figure can be found in S2 Data.

of 21 human neurons and 6 of 16 mouse neurons, aligning with results from the Allen Institute open access database, which reports markedly low or absence of average *PVALB* mRNA expression levels in some layer 2/3 pvalb cells (Fig 8B). Natural fluctuations in different gene expression levels in individual cells might explain the occasional undetectable mRNA in our patch-sequenced cells, but it is also possible that our system occasionally loses mRNA or misses its conversion to cDNA for reasons we do not yet recognize.

The patch-sequencing results closely aligned with the average mRNA profiles of pvalb neuron types in the Allen Institute database (Fig 8B). The database confirmed the absence of *KCNA1* expression in humans but not in mice ($p < 0.001$), as well as expression of *KCNA2* in both species. It also confirmed that among the potassium channel genes analyzed, only Kv1 channel genes were consistently expressed at lower levels in human pvalb neurons relative to mouse neurons (one-way MANOVA with a *post hoc* Bonferroni test).

The heatmaps based on our patch-sequencing data and the Allen Institute data differ in that our dataset shows raw TPM results in individual neurons, whereas the Allen Institute data show the average TPM level for multiple cells of that cell type. This may explain why, unlike the Allen Institute dataset, ours occasionally shows undetectable mRNA levels for common genes, such as *SLC6A1*, in some cells. It is also noteworthy that the Allen Institute dataset shows an absence of mRNA levels for *PVALB* in some cell types, even though the dataset represents the average of many cells.

## Kv1 channel blocker dendrotoxin lowers the AP firing threshold more prominently in mouse neurons than in human neurons

We investigated whether Kv1 channel blockade through dendrotoxin-K (DTXK) affects AP generation differently in human and mouse pvalb neurons [53,58]. After 5 min of baseline whole-cell current–clamp recording (conditions similar to those shown in Fig 1), slices were exposed to 100 nM DTXK for 10 min, and AP threshold was remeasured (Fig 9A₁ and 9A₂). DTXK significantly altered the AP firing threshold in human neurons from −41.5 mV (IQR = −44.0 to −39.0 mV) to −41.0 mV (IQR = −45.3 to −41.0 mV; $n = 11$ cells; $p = 0.0459$; mixed-design ANOVA and post hoc Wilcoxon test with Holm's correction). In mouse cells, the AP threshold was lowered from −32.0 mV (IQR = −35.5 to −28.5 mV) to −39.3 mV (IQR = −44.2 to −38.0 mV; $n = 15$ cells; $p = 0.0007$; mixed-design ANOVA and post hoc Wilcoxon test with Holm's correction). The change in AP threshold between baseline and in DTXK differed significantly between the human and mouse cells ($p = 0.0009$ for species interaction by mixed-design ANOVA) (Fig 9B₁–9B₃). The AP threshold was more negative in humans than in mice at baseline ($p = 0,0005$), but there was no statistically significant difference between the species at DTXK ($p = 0.0862$) (mixed-design ANOVA and post hoc Mann–Whitney *U* tests between species at each time point, with Holm-adjusted p-values). Table 2 show the full statistical analysis and its explanation.

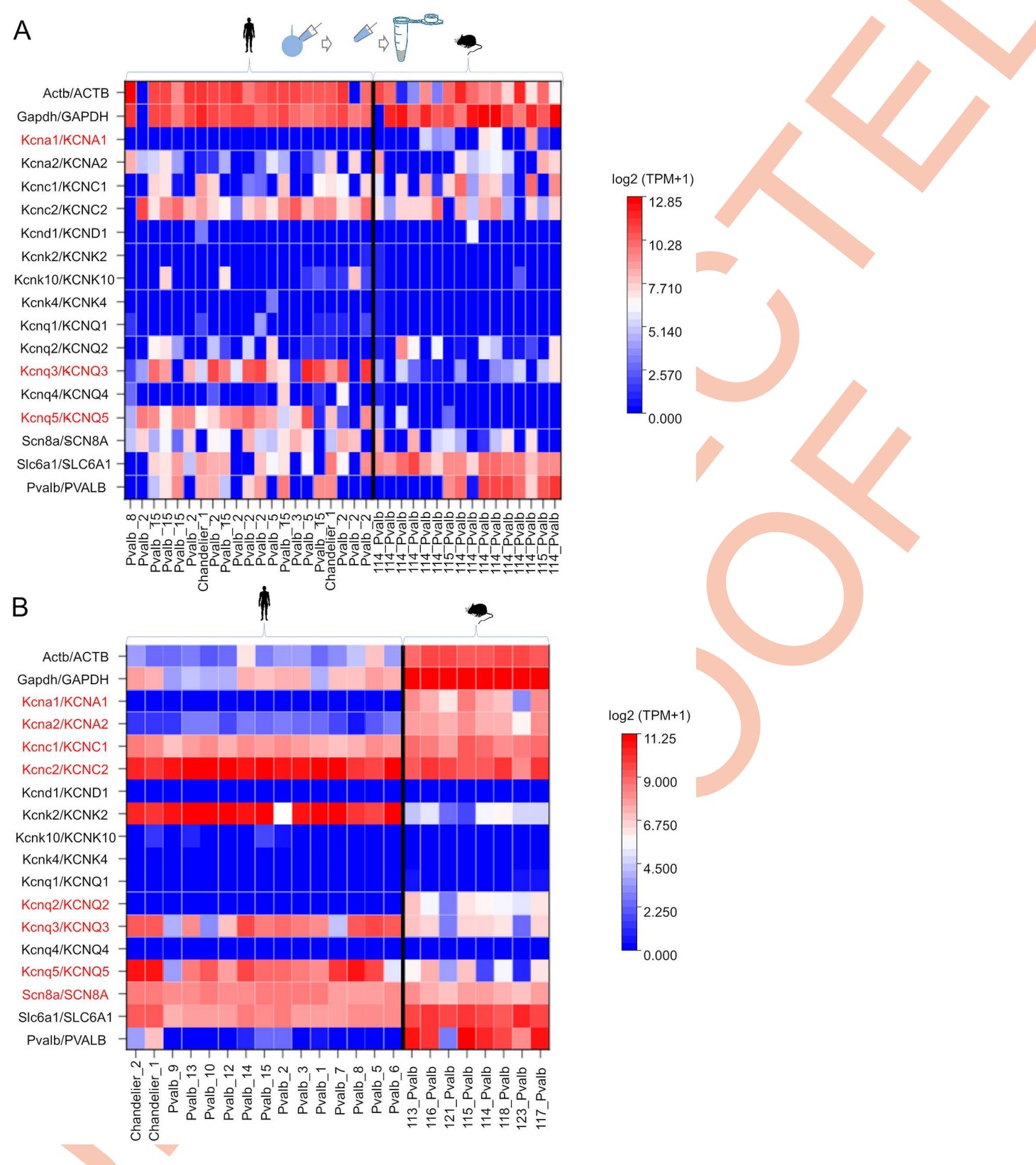

**Fig 8. Human pvalb neurons lack *KCNA1* mRNA and express low *KCNA2* mRNA levels. (A)** Patch-sequencing analysis (illustrated in the upper schematic) of individual neurons revealed that human pvalb neurons do not express *KCNA1*, in contrast to its expression in mouse cells. **($A_1$)** Heatmap showing transcript levels (TPM) of genes encoding low-threshold potassium channels in individual human ($n = 21$) and mouse ($n = 16$) pvalb neurons, as well as the housekeeping genes *GAPDH* and *ACTB*. *KCNA1* expression differed quantitatively (TPM values) and categorically (positive or negative gene expression) in human and mouse cells. Expression of *KCNA2*, *KCNC1*, *KCNC2*, *KCNK2*, *KCNK4*, and *KCNK10* was observed in both species (*61*), whereas *KCND1* expression was nearly absent in both (*62*). 0-level indicates that the mRNA expression level is below the detection threshold. Human and mouse cells expressed similar levels of *KCNQ1*, *KCNQ2*, and *KCQN4*, whereas *KCNQ3* and *KCNQ5* expression levels were higher in humans. Additionally, *SCN8A* (encoding Nav1.6 channels) and *SLC6A1* (encoding GABA transporter type-1) were highly expressed in both species. Significant differences in TPM values are marked in red (genes, ordinate). Bottom of the heatmap shows the pvalb cell subtype classification based on Allen Institute transcriptomic data. **(B)** Allen Institute database analysis of layer 2/3 pvalb neuron subtypes in the human and mouse neocortex confirmed *KCNA1* mRNA deficiency in human cells. Heatmap shows median TPM values (scale on the right) across seven human and eight mouse pvalb neuron subtypes (abscissa). *KCNA1* mRNA was absent in all human subtypes but robustly expressed in mouse cells. Statistical analyses (one-way MANOVA with Bonferroni *post hoc* test and chi-squared test with Benjamini–Hochberg adjustment) confirmed significant differences in both expression level and frequency ($p = 0.0011$ and $p < 0.001$, respectively). *KCNQ3* and *KCNQ5* expression levels were higher in humans. Significant differences in TPM values are marked in red (genes, ordinate). Abscissa shows pvalb cell subtype names. The data underlying this Figure can be found in S3 Data.

Comparisons of the effect of DTXK across species revealed that, in addition to an AP threshold shift, the rheobase current was affected differently in mice than in humans (mixed-design ANOVA interaction, $p = 0.0082$). The rheobase current was reduced in mice ($p = 0.0015$), but not in humans ($p = 0.4648$; post hoc Wilcoxon tests with Holm's correction) (Fig $9C_1$–$9C_3$). Although the rest of the observed parameters failed to reveal a significant difference in the DTXK effect between humans and mice (mixed-design ANOVA interaction, $p > 0.05$), Kv1 channel blockade was associated with a reduction in the amplitude of the action potential afterhyperpolarization (AP AHP) in mice, but not in human cells. This difference reflects the stronger somatic Kv1 channel contribution to AP AHP in mice compared to human cells in control conditions. The AP half-width, which was shorter in mice under baseline conditions, did not remain significantly shorter after the wash-in of DTXK, reflecting the blockade of K-current components of the AP in mice. Additionally, DTXK was associated with a similar reduction in the maximum positive peak value of the AP in both species. This suggests that the effect of DTXK on the AP occurred through a mechanism other than direct Kv1.1 channel current. (A mixed-design ANOVA test with post hoc pairwise comparisons are shown in full in Table 2.) In addition, DTXK significantly increased input resistance in mouse cells but not in human cells. Kv1.1 channels are activated in the cell soma by $V_m$ depolarization, which reduces Rin under baseline control conditions. Blocking these channels with DTXK increased the input resistance. Consistent with this finding, DTXK reduced species differences in membrane "tau". "Tau" differed more between species in control conditions than in the presence of DTXK (see Table 2). The results indicate stronger Kv1 channel activity in the AIS and soma of mouse neurons compared to human pvalb neurons.

## Fast-spiking neuron computational model with a human-type AIS reveals a lowered firing threshold and shortened time lag to AP generation

We established a single-cell computational model with axonal and dendritic cable properties and an AIS replicating the electrical characteristics of human pvalb neurons. We simulated the effects of AIS features that differed between species—namely, AIS length and Kv1 channel content—on AP firing threshold and the time lag between somatic excitation and AP initiation. The model's intrinsic properties were tuned to match the $V_m$ waveforms and AP firing threshold observed in human pvalb cells during voltage step protocols (Fig $10A_1$ and $10A_2$). Specifically, the model reproduced a whole-cell recorded human pvalb cell (code h7, S1 Table) with Hodgin–Huxley-type APs and incorporated voltage-sensitive low-threshold Na$^+$ current (Nav1.6 in the AIS), high-threshold Na$^+$ current (Nav1.2 in the soma and proximal dendrites), and K$^+$ currents (Kv3.1 and Kv3.2 in the AIS, soma, and dendrites). The soma and dendrites also included voltage-insensitive K$^+$ leak current, M-type and Kir-type K$^+$ currents [63], and hyperpolarization-activated nonspecific cation Ih current [28]. Variable Kv1 activity (conductance) was applied specifically in the AIS. Under standard conditions, the AIS length was 20 µm, initiation distance from the soma was 10 µm, and the AP firing threshold was −43 mV.

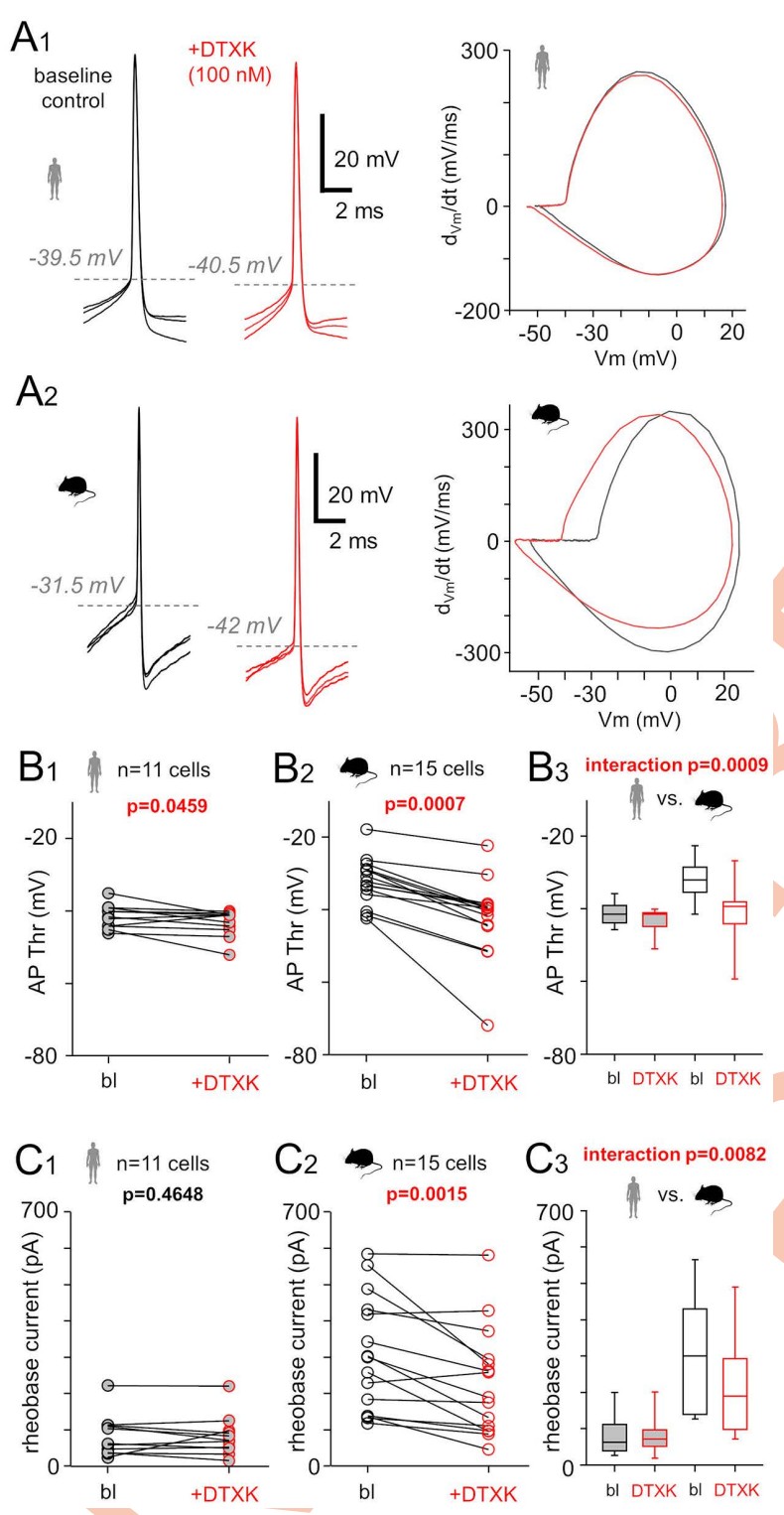

**Fig 9. Kv1 channel blocker dendrotoxin lowers AP threshold in mouse but not in human pvalb neurons.** APs were recorded under baseline conditions (black traces) in current–clamp mode at −70 mV (as shown in Fig 1) for 5 min, followed by application of 100 nM DTXK for 10 min, after which APs were evoked again (red traces). **(A)** APs and corresponding AP phase plots in baseline control and after DTXK application. **(A₁)** Left: Three super-imposed APs from a human pvalb cell. Right: Corresponding phase plots under control and DTXK conditions. **(A₂)** Left: Superimposed APs recorded in a

mouse pvalb neuron. Right: Corresponding AP phase plots. **(B–C)** The AP parameters that were affected differently by DTXK in human and mouse pvalb neurons (mixed-design ANOVA). **(B)** Effect of DTXK on the AP threshold in human (**B**$_1$) and mouse (**B**$_2$) neurons (Wilcoxon signed-rank tests within each species with Holm-adjusted *p*-values for multiple comparisons). (**B**$_3$) The mixed-design ANOVA indicates a significant interaction between species and time on the measured values, meaning the DTXK treatment differs between human and mouse neurons. Box plots present the median, quartiles, and 5th and 95th percentiles. **(C)** Application of DTXK did not affect the rheobase current in human pvalb neurons (**C**$_1$), but it significantly reduced the current in mouse cells (**C**$_2$) (Wilcoxon signed-rank test with Holm-adjusted p-values for multiple comparisons). (**C**$_3$) A significant interaction was found between species and time in the rheobase values, indicating that DTXK treatment differs between humans and mice (mixed-design ANOVA). The data underlying this Figure can be found in S4 Data.

First, we simulated the AP firing thresholds by varying AIS lengths (10–30 μm), within the anatomical range observed in real pvalb neurons of the two species (see Fig 3A). Increased AIS length linearly scaled Nav1.6 and Kv1 channel activity in the AIS (conductances: gNav1.6 and gKv.1 = 20 nS/μm$^2$). The axon diameter was 0.8 μm, with a longitudinal resistance (Ra) of 0.9 MΩ·μm. Increasing AIS length from 10 to 20 and 30 μm lowered the AP firing threshold (elicited by a 250-ms square-pulse current at the soma of −70 mV) from −36.8 to −42.3 and −45.2 mV, respectively (Fig 10B$_1$ and 10B$_2$), demonstrating that AIS length (with fixed gNav1.6 and gKv.1 per μm$^2$) strongly modulates AP threshold in individual neuron.

Next, we examined how varying Kv1 conductance in the AIS affects AP threshold while fixing the AIS length (20 μm), location along the axon (10 μm from the soma, Ra = 0.9 MΩ·μm), and gNav1.6 density (20 nS/μm$^2$). Depolarizing steps were applied to elicit APs. Increasing gKv1 from 0 to 5, 10, and 15 nS/μm$^2$ in the AIS (Fig 10C$_1$ and 10C$_2$) raised the AP threshold from −45.3 to −44.2, −42.4, and −41.6 mV, respectively (Fig 10D). This simulation confirmed that Kv1 conductance in the AIS increases firing threshold independently of AIS geometry. We also simulated increased Nav1.6 conductance in the AIS, and found that it effectively reduced AP threshold in the model cell (S1A Fig).

We then ran simulations incorporating cell-to-cell variability parameters shared across species. With fixed AIS length (20 μm) and channel conductances (gNav1.6 and gKv1 = 20 nS/μm$^2$), we varied the AIS initiation site distance (0 or 10 μm from the soma). AP threshold was lower when AIS was 10 μm from the soma (−43.2 mV) compared with 0 μm (−41.5 mV; S1B and S1C Fig). We also simulated the impact of axon diameter variation (0.6–0.9 μm), which alters both the Ra (0.8–1.1 MΩ·μm) and extracellular membrane surface area (calculated as diameter × π, with gNav1.6 and gKv1 = 20 nS/μm$^2$). Narrower axons (0.6 μm) with higher Ra (1.1 MΩ·μm) yielded the lowest AP threshold (−44.7 mV) in the model cell, whereas wider axons (0.9 μm) with lower Ra (0.8 MΩ·μm) had the highest threshold (−42.2 mV). Varying Ra alone (with a fixed axon surface area) shifted the threshold by <1 mV; changing surface area alone (while keeping Ra constant), thereby increasing total ion channel conductance, altered the threshold by ~2 mV. Notably, when axon diameter was increased over the range observed in the anatomical study, thereby lowering Ra while increasing total Nav1.6 and Kv1 channel conductance, the AP firing threshold moderately increased only by ~2.5 mV. These relationships are detailed in the heatmap shown in S1B and S1C Fig.

Finally, we used the model cell to explore how AIS-dependent changes in firing threshold influence the time lag between EPSC onset AP generation in pvalb neurons. The simulation aimed to address why human pvalb neurons, compared with mouse pvalb neurons, despite having a slower somatic membrane "tau" (Fig 10E$_1$) and a lower AP firing threshold (Fig 10E$_2$), can generate APs with an equal or shorter time lag in response to a fixed-amplitude excitatory current in the soma (Fig 10E$_3$ and 10E$_4$) [38]. In the model neuron, we generated an EPSC in the soma at −70 mV and incrementally increased its peak amplitude to evoke a gradually increasing EPSP that eventually reached the AP firing threshold [28]. The EPSC was applied with peak conductance values from 6 to 11 nS in steps of 0.25 nS, reaching the AP firing threshold with the human- and mouse-type AIS configurations at EPSC conductances of 6.75 and 8.50 nS, respectively (Fig 10F$_1$–10F$_3$). Shifting from the "mouse-type AIS" (Kv1 density = 20 nS/μm$^2$) and short AIS length (AIS length = 10 μm) to the "human-type AIS" (Kv1 density = 0 nS/μm$^2$) and long AIS (AIS length = 30 μm), which showed a maximal shift

**Table 2. AP parameters, Rin, and "tau" in pvalb neurons under baseline and DTXK exposure conditions.**

| | Human n = 11 | | Mouse n = 15 | | P-value | | | | |
|---|---|---|---|---|---|---|---|---|---|
| | Baseline median, (IQR) | DTX, median, (IQR) | Baseline median, (IQR) | DTX, median, (IQR) | Mixed design ANOVA (species, time, interaction) | Human (from baseline to DTX) | Mouse (from baseline to DTX) | Between human and mouse in baseline | Between human and mouse (in DTX) |
| AP Thr (mV) | −41.5 (−44.0 to −39.0) | −41.0 (−45.30 to −41.0) | −32.0 (−35.5 to −28.5) | −39.3 (−44.2 to −38.0) | **0,0348** **1,64E−06** **0,0009** | **0,0459** | **0,0007** | **0,0005** | 0,0862 |
| Initial rise slope (ms⁻¹) | 41.1 (33.0 to 44.7) | 43.4 (30.5 to 81.6) | 45.2 (38.6 to 61.7) | 45.3 (41.0 to 65.7) | 0,5107 0,2773 0,9664 | – | – | – | – |
| AP ampl (peak positive value, mV) | 19.0 (10.7 to 26.6) | 13.4 (3.8 to 17.7) | 20.1 (15.5 to 24.2) | 11.1 (5.0 to 20.7) | 0,8017 **0,00002** 0,6541 | **0,0010** | **0,00006** | 1.0000 | 1.0000 |
| AP half-width (ms) | 0.54 (0.44 to 0.90) | 0.52 (0.40 to 0.80) | 0.34 (0.33 to 0.43) | 0.38 (0.34 to 0.49) | **0,0173** 0,6613 0,3315 | 0,3223 | 0,3028 | **0,0127** | 0,1193 |
| AP rheobase (pA) | 62.3 (37.5 to 113.0) | 71.4 (50.2 to 97.8) | 298.0 (138.4 to 430.2) | 188.9 (97.4 to 294.0) | **0,0006** **0,0019** **0,0082** | 0,4648 | **0,0015** | **0,00006** | **0,0026** |
| AP latency (ms) | 23.2 (20.6 to 27.6) | 25.9 (17.5 to 48.9) | 62.3 (38.2 to 109.1) | 32.8 (15.4 to 69.1) | 0,1028 0,2588 0,5471 | – | – | – | – |
| Spike number by step | 3.3 (1.0 to 7.5) | 2.3 (2.0 to 5.3) | 5.0 (3.3 to 10.4) | 7.5 (2.5 to 11.0) | 0,1492 0,9571 0,5136 | – | – | – | – |
| AP AHP ampl (mV) | −12.7 (−7.4 to −18.1) | −10.5 (−8.4 to −17.2) | −17.9 (−15.5 to −20.8) | −13.7 (−8.2 to −15.8) | **0,0449** 0,0823 0,4536 | 0,4131 | **0,00006** | 0,7164 | 0,1461 |
| Max rise speed (mV/ms) | 208.5 (122.1 to 315.3) | 211.7 (63.6 to 341.4) | 289.2 (245.4 to 368.6) | 355.6 (151.4 to 417.7) | 0,1843 0,1650 0,7324 | – | – | – | – |
| Max fall speed (mV/ms) | −135.8 (−161.7 to −101.1) | −132.8 (−159.1 to −96.5) | −169.6 (−238.3 to −117.2) | −129.7 (−240.7 to −97.6) | 0,3005 0,0948 0,5920 | – | – | – | – |
| Rin (MΩ) | 239.2 (103.7 to 356.8) | 253.9 (124.5 to 419.5) | 84.6 (77.3 to 104.1) | 102.0 (84.7 to 120.6) | **0,0043** **0,0139** 0,9423 | 0,1475 | **0.0026** | **0,0147** | **0,0095** |
| "Tau" (ms) | 7.2 (4.9 to 8.6) | 7.4 (5.4 to 8.3) | 4.2 (3.8 to 5.3) | 4.6 (3.6 to 6.7) | **0,0119** 0,5922 0,1514 | 0,1354 | 0,5772 | **0,011** | 0,0868 |

Rin: cell input resistance; "tau": apparent membrane time constant, measured as the voltage response to a +10-mV step from −70 mV. Data are presented as medians with quartiles. Mixed-design ANOVA $p < 0.05$ values demonstrate a statistically significant main effect of species, indicating an overall difference in measurement values between human and mouse cells, averaged across time points. A main effect of time ($p < 0.05$) demonstrates a significant change in measurement values from baseline to 10 min after DTXK, averaged across species. The interaction effect ($p < 0.05$) between species and time indicates that the change in measurement values over time differs significantly between human and mouse cells. When the mixed-design ANOVA values showed $p < 0.05$, post hoc tests were applied to compare the baseline and DTXK values within each species (Wilcoxon signed-rank test with Holm-adjusted $p$-values for multiple comparisons), or at specific time points between species (Mann–Whitney $U$ tests with Holm-adjusted $p$-values). The data underlying this Table can be found in S4 Data.

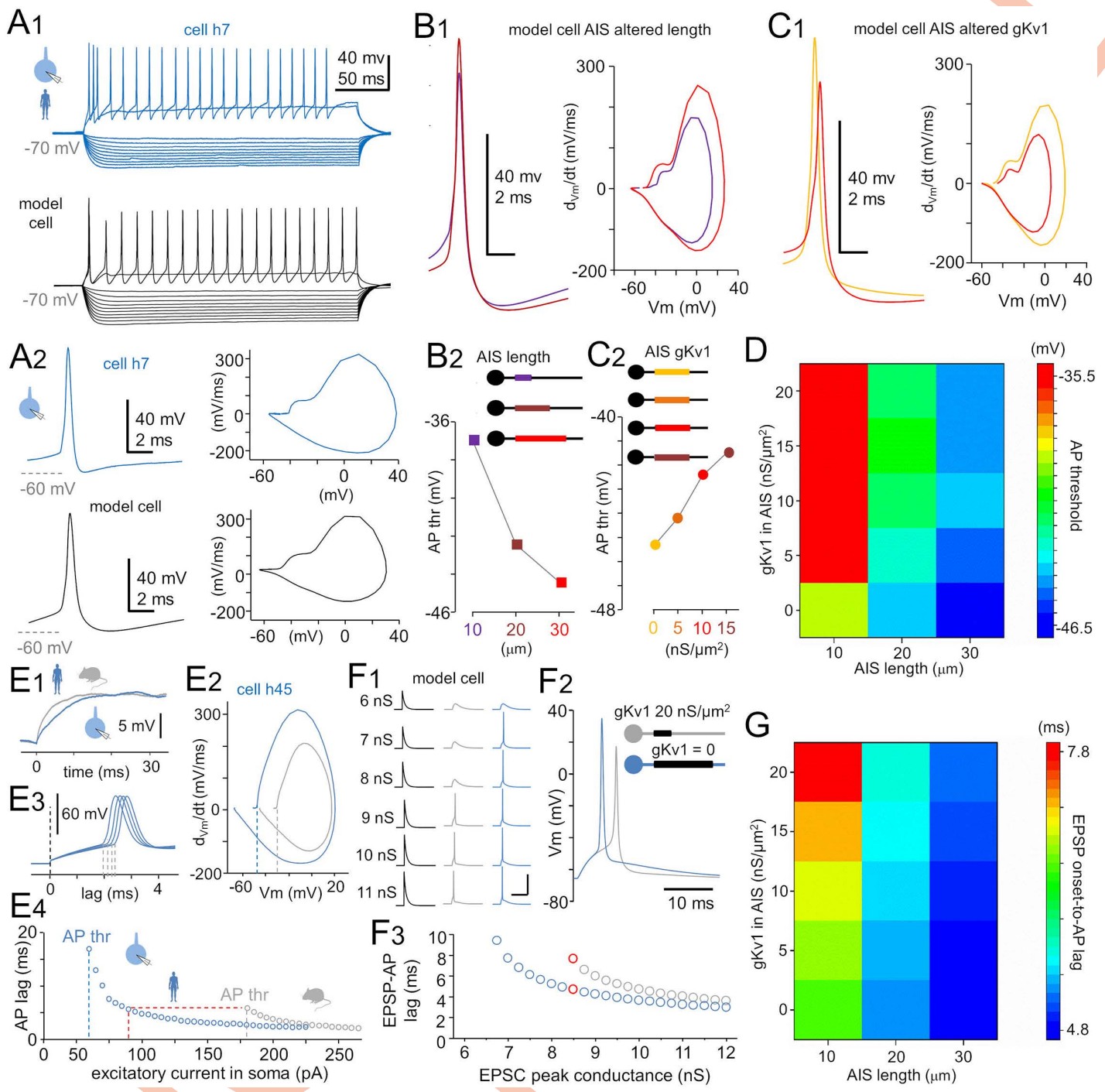

**Fig 10. Fast-spiking neuron computational model with human or mouse AIS. (A)** Single-cell computational model with axon and dendrite cable properties replicating the human pvalb cell's somatic electrical behavior. **(A₁)** $V_m$ responses in a real human pvalb neuron (blue) and the model cell (gray) to current steps from −200 to +200 pA. **(A₂)** Left: First AP elicited by the lowest suprathreshold step in the real neuron (blue) and model cell (gray). Right: Corresponding AP phase plots. **(B)** Increasing AIS length from 10 to 30 μm, within the observed range in real pvalb cells of the two species (see Fig 3A₂), lowered the AP firing threshold in the model (Kv1 and Nav1.6 channel conductance = 20 nS/μm², AIS-to-soma distance = 10 μm, axon Ra = 0.9 MΩ·μm, axon diameter = 0.8 μm). **(B₁)** Model APs and phase plots with AIS lengths of 10 μm (purple) and 30 μm (red). **(B₂)** AP firing threshold simulated for AIS lengths of 10 μm (purple), 20 μm (brown), and 30 μm (red) (illustrated in schematic inset). Threshold sensitivity was highest at 10–20 μm, a range common to both human and mouse pvalb neurons in reality. **(C)** Reducing Kv1 channel activity in the AIS lowered the AP firing threshold in

the model. **(C₁)** Model APs and phase plots at different Kv1 conductances in the AIS: gKv1 = 0 (yellow) and 15 nS/µm² (red). **(C₂)** AP firing threshold at Kv1 conductance levels of 0, 5, 10, and 15 nS/µm² (abscissa; Nav1.6 conductance = 20 nS/µm², AIS length = 20 µm, AIS-to-soma distance = 10 µm). **(D)** Combined AIS elongation and reduced Kv1 activity synergistically lowered the AP threshold in the model. Heatmap illustrates the combined effect of AIS length (abscissa) and Kv1 conductance (ordinate) on the AP firing threshold (color scale, right). Total AP threshold range across conditions was 11 mV. **(E–G)** Human-type AIS shortened the AP generation lag. **(E)** Real human and mouse neurons differed in depolarization kinetics and AP thresholds but showed similar AP generation lags. Human pvalb neuron (blue) had a slower membrane depolarization **(E₁)** and lower AP threshold **(E₂)** relative to the mouse pvalb neuron (gray). **(E₃)** AP generation lag, defined from the onset of a depolarizing step (black dotted lines) to AP initiation (gray dotted lines) in four representative traces (AP onset defined at 10-mV/ms rise speed). **(E₄)** AP generation lag vs. excitatory current steps in the soma of human (blue) and mouse (gray) neurons. Human neuron fired with a lower current (abscissa; vertical dotted lines) but showed a similar time lag (horizontal dotted line) relative to the mouse neuron. **(F)** Switching the model's AIS properties from mouse-like (high Kv1 content) to human-like (no Kv1) while simultaneous changing the AIS length from short to long reduced the AP generation lag to somatic excitation. **(F₁)** The AIS without Kv1 and increased length lowered the AP threshold in response to simulated EPSCs in the model neuron soma. $V_m$ traces showed excitatory postsynaptic potentials (EPSP) of increasing strength that eventually triggered an AP in model neurons with human (blue) and mouse (gray) AIS. EPSP-generating conductances with peak amplitudes of 6–nS are presented on the left (black traces). Scale bars: 40 ms and 50 mV. **(F₂)** The AIS without Kv1 and increased length shortened the AP generation lag in response to EPSPs. $V_m$ traces showed APs triggered by equal-strength excitatory postsynaptic currents (EPSC) (8.5-nS peak conductance) in model neurons with long AIS without Kv1 (blue) or short AIS with high Kv1 content (gray). **(F₃)** The plot shows AP generation lag against increasing EPSC conductance in model neurons with the two types of AIS (encoded by blue and gray dots, respectively). Response to the 8.5-nS EPSC is highlighted in red. **(G)** Heatmap showing the combined effect of AIS length (abscissa) and AIS Kv1 conductance (ordinate) on the AP generation time lag in response to a fixed-strength EPSC (8.5 nS) in the model neuron. Color scale (right) indicates time lag in milliseconds.

in the AP Thr (Fig 10D), resulted in a shorter time lag to AP generation when the EPSC strength was fixed (Fig 10F₂ and 10F₃). The EPSP–AP time lag simulation with varying AIS lengths and Kv1 densities is shown in a heatmap in Fig 10G. The simulation revealed a range of AP generation time lags from 4.8 and 7.8 ms in response to a fixed EPSC strength (8.5 nS, corresponding to the AP threshold in the mouse-type AIS). Overall, these simulations demonstrate that lower AP firing thresholds, generated by AIS parameters that differ between human and mouse real pvalb neurons, are accompanied by shorter time lags to AP generation in response to synaptic currents at the soma. Importantly, the simulations show that variability in Kv1 content alone in the AIS is sufficient to regulate AP threshold (ordinate values in Fig 10D) and EPSP onset-to-AP lag (ordinate values in Fig 10G).

## Discussion

Species-specific traits in organs, tissues, and cells have evolved in response to selection pressure during phylogenetic evolution [8,64,65]. Consequently, homologous cell types (i.e., neurons expressing the same cell-type specific marker genes) are broadly conserved across mammalian species but still exhibit interspecies differences that reflect adaptations for functional advantages [4,66–69]. Our findings suggest that fast-spiking pvalb neurons in the human neocortex evolved a lower AP firing threshold to support their role as fast in–fast out circuits. This enhanced input-to-output signal transformation speed is especially important in human neurons, which have electrically slower somata compared with rodent neurons. Molecular, anatomical, and pharmacological data, together with model simulations, showed that the reduced AP threshold in human pvalb neurons results predominantly from diminished Kv1 inhibitory potassium channel content in the AIS.

### Low AP firing threshold helps electrically slower human interneurons to function as fast in–fast out circuits

A short delay in AP generation in response to excitatory synaptic input (i.e., input–output transformation) is a key feature of fast-spiking pvalb interneurons in the cortex across mammals [40,43]. These neurons, the most numerous inhibitory population in the cortex [6,70,71], exert fast, temporally precise inhibition on various other neuron types [43]. In the human neocortex, rapid signal transformation poses a challenge for pvalb neurons owing to their slower somatic membrane potential kinetics. However, a lowered AP threshold compensates for this, reducing the time lag in input–output transformation. This helps preserve the archetypical fast in–fast out function of pvalb neurons in humans, as demonstrated through our computational model.

Slowed inhibitory signaling in the human neocortex would be detrimental, as theoretical models and in vivo animal studies have linked such changes to reduced cortical computation speed and cognitive impairment [40,72–75]. Therefore, human fast-spiking interneurons have evolved distinct adaptations that maintain rapid electrical input–output transformations. These include simplified dendritic morphology that facilitates synaptic potential propagation from distal dendrites to the soma [2,31]. Moreover, human pvalb neurons express more voltage-sensitive excitatory cation channels in the soma and proximal dendrites to accelerate membrane potential dynamics [28]. We also observed an adaptation in AP generation at the soma. Some of these features likely generalize to primate pvalb neurons, as macaque neocortical neurons also exhibit slower $V_m$ kinetics and reduced AP thresholds compared with rat neurons [36].

### Why do human pvalb neurons have electrically slower somata relative to rodent neurons?

The slower soma membrane dynamics in human pvalb neurons result from increased cell Rin, which reflects reduced transmembrane ion leakage. This is supported by findings showing that cell Cm, a parameter that combines with Rin to determine apparent membrane tau, is similar between primate and rodent fast-spiking neurons [36,37]. In mice, somatic "tau" correlates with Rin, but in humans, this link is weak, indicating high variability in membrane ion channel density between individual human neurons (hence, in electrical resistivity per plasma membrane area), which governs ion leakage [76].

The reason high Rin evolved in human pvalb neurons remains unclear, as it is inherently unfavorable for fast in–fast out function. One hypothesis is that evolutionary pressure favored neurons that produce $V_m$ changes with reduced electric current demands, minimizing transmembrane ion flux to conserve metabolic energy [77,78]. Due to the lower AP threshold and higher input resistance, human cells require less current to reach the spike threshold. Fast-spiking interneurons are energetically demanding [79–81], and their metabolic cost may be heightened in the human brain due to the higher proportion of interneurons in the cortex compared with that in rodents [14,82,83]. However, the increased electrical resistivity of the plasma membrane in human pvalb neurons likely necessitated an adjustment of the AP threshold to preserve their rapid electrical input–output transformation.

### Mechanisms underlying the lowered firing threshold in human fast-spiking neurons

Experiments using a Kv1 channel blocker directly demonstrated the predominant role of Kv1 channels in setting a higher AP threshold in mouse pvalb neurons compared with human pvalb neurons. Immunofluorescence analysis revealed a deficiency of Kv1.1 channels in human neocortical pvalb neurons, and *KCNA1* expression was absent in all human pvalb neurons, despite being present in mouse cells. These findings aligned with results from the Allen Institute transcriptomic database on averaged mRNA levels in human and mouse pvalb neuron subtypes. Although Kv1.2 channels were similarly expressed in both species, immunostaining showed weak AIS labeling in each, and the Allen Institute dataset indicated lower average Kv1.2 levels in human-type compared with mouse-type pvalb cells. However, the combined mRNA levels of Kv1.1 and Kv1.2 did not correlate with AP threshold across individual cells. Larger datasets in future studies may help clarify this relationship. Both datasets showed that the Kv1 channel gene group was the only potassium channel group expressed at lower levels in human pvalb neurons compared with mouse neurons. We argue that species differences in Kv1.1 channel density are the major determinant of the lower AP threshold in humans. Although this work does not determine the exact AP initiation site in the AIS, it shows that the low-threshold ion channels Nav1.6 and Kv1 are, on average, evenly distributed along the AIS, without a consistent intensity peak at either end.

Human pvalb neurons have a moderately longer AIS on average relative to their mouse counterparts, and simulations showed that AIS length modulates AP threshold within the physiological range observed. Although AIS length contributes to AP threshold variability among individual pvalb cells in both species, it does not explain the interspecies difference. While an increased AIS length increases the number of Na$^+$ channels in the AIS, making it easier to reach the AP threshold, as demonstrated in S1 Fig, an elongated AIS does not necessarily generate differences in AP thresholds between

species. The spike threshold is influenced by various factors, including axial resistance between the soma and AIS, as well as electrical resistance within the AIS. Possible species differences in these parameters may mask the effect of moderately elongated AIS on AP thresholds in human neurons.

AIS elongation lowers the AP threshold by increasing the total number of Na channels in the initial segment [41,48]. Increasing AIS length reduces AP threshold more effectively than expanding axon diameter, although both increase the number of AIS ion channels. However, expanding axon diameter also lowers Ra, which shifts AP threshold in the opposite direction and limits its effect [48]. In rodents, AIS elongation coincides with changes in voltage-gated potassium channel composition to enhance excitability [45]. Human pvalb neurons similarly exploit both geometry and reduced Kv1 activity to regulate AP threshold.

## Mechanisms underlying cell-to-cell variability in the AP threshold without interspecies differences

Additionally, the distance from the AIS initiation site to the soma, the axon diameter, and the axon Nav1.6 channel immunofluorescence intensity vary among individual neurons, yet these factors do not explain the species-specific differences in AP threshold. However, these features likely contribute to cell-to-cell variability in pvalb neuron AP thresholds in both species, as indicated by our model cell simulations.

Increases in axon diameter [48] typically occur with reduced Ra, which raises the AP threshold in pvalb interneurons. Given the evolutionary pressure to lower the AP threshold in human neurons, it follows that axon diameter in human pvalb neurons is not markedly larger than that in mouse cells, as enlargement would counteract AP threshold reduction.

The AIS is not a static cellular microdomain [84], and the observed cell-to-cell variability in AIS properties likely results from plasticity mechanisms [85], enabling the brain to adapt, learn, and repair. Whether these mechanisms differ between humans and animal models remains unknown.

## Conclusions

Results and conclusions regarding neocortical neural networks are likely influenced by species-specific neuronal features. Studies that rely solely on animal models may provide an inaccurate picture of human brain network physiology and aging. The same concerns apply to degenerative mechanisms underlying disease processes. Therefore, identifying human-specific neural features is crucial to developing effective therapeutic strategies for degenerative brain conditions, in which pvalb neurons are often emphasized.

## Study limitations

The lack of biotechnological tools to selectively and rapidly manipulate gene expression, and consequently the AIS ion channel composition and microstructure, in genetically defined neuron types, such as pvalb neurons, in human brain slices continues to constrain experimental design. In addition, immunostaining-based localization and quantification of proteins are limited by antibody validation, which currently relies on knockouts only feasible in nonhuman species. Future availability of such techniques in human tissue will enable a more detailed understanding of the human-type AIS in health and disease.

## Materials and methods

### Ethics statement

Written informed consent was obtained from all patients before surgery. For underage patients, consent was obtained from a parent or guardian.

*Licenses:* All procedures involving animals were approved by the Governmental Office of Animal Wealth and Welfare Department (permit number CS/I01/03036-2/2024) and the University of Szeged Ethics Committee. Human studies were

approved by the Regional Human Investigation Review Board (reference number 75/2014) and the National Scientific and Research Ethics Committee (ETT TUKEB) (license number BM/25042-1/2024). All studies were conducted in accordance with the tenets of the Declaration of Helsinki.

## Human brain slices

Neocortical slices were prepared from frontal, temporal, or other cortical samples resected during deep-brain surgeries. Patients ranged in age from 11 to 85 years, and samples were obtained from both hemispheres of male and female patients (S1 Table). Anesthesia was induced with intravenous midazolam (0.03 mg/kg) and fentanyl (1–2 μg/kg) after a bolus intravenous injection of propofol (1–2 mg/kg). Rocuronium (0.5 mg/kg) was administered to facilitate endotracheal intubation. During surgery, patients were ventilated with a 1:2 $O_2$/$N_2O$ mixture, and anesthesia was maintained with sevoflurane. After resection, tissue blocks were immediately immersed in an ice-cold solution containing (in mM) 130 NaCl, 3.5 KCl, 1 $NaH_2PO_4$, 24 $NaHCO_3$, 1 $CaCl_2$, 3 $MgSO_4$, and 10 d-(+)-glucose, which was aerated with 95% $O_2$/5% $CO_2$ in a glass container. The container was kept on ice in an insulated box and transported within 20 min to the electrophysiology laboratory under continuous aeration (95% $O_2$/5% $CO_2$). Slices (350-μm thick) were cut using a vibrating blade microtome (Microm HM 650 V, Thermo Fisher Scientific, Cambridge, UK) and incubated for 1 h at 22–24 °C in slicing solution. This solution was gradually replaced by recording solution (180 mL) delivered at 6 mL/min by pump. The recording solution matched the slicing solution except for 3 mM $CaCl_2$ and 1 mM $MgSO_4$.

## Mouse brain slices

Transverse slices (350 μm) were prepared from the somatosensory ($n = 45$) and frontal ($n = 50$) cortices of 5–7-week-old heterozygous B6.129P2-Pvalbtm1(cre)Arbr/J mice (stock 017320, B6 PVcre line; Jackson Laboratory, Bar Harbor, ME, USA) crossed with Ai9 reporter mice to express the tdTomato fluorophore in pvalb GABAergic neurons, which aided cell selection. Cells were tentatively identified electrophysiologically through fast spike kinetics and high-frequency, nonaccommodating firing in response to 500-ms suprathreshold depolarizing pulses [38].

## Electrophysiological recordings and data analysis

Recordings were performed in a submerged chamber perfused at 8 mL/min with recording solution maintained at 36–37 °C. Cells were patched under visual guidance using infrared differential interference contrast video microscopy with a water-immersion ×20 objective, additional zoom (up to ×4), and a dichroic mirror and filter (Cy3) for TdTomato fluorescence in mouse tissue. All recordings were conducted within 30 min of entering whole-cell mode. Micropipettes (5–8 MΩ) were filled with intracellular solution containing (in mM) 126 K-gluconate, 8 NaCl, 4 ATP-Mg, 0.3 $Na_2$-GTP, 10 HEPES, and 10 phosphocreatine (pH 7.0–7.2; 300 mOsm) supplemented with 0.3% (w/v) biocytin for *post hoc* staining with fluorophore-conjugated streptavidin. Recordings were performed with a Multiclamp 700B amplifier (Axon Instruments, Scottsdale, AZ, USA) and low-pass-filtered at 6–8 kHz (Bessel filter). Series resistance and pipette capacitance were compensated in current–clamp mode. $V_m$ values were not corrected for the liquid junction potential. Somatic apparent tau was measured in current–clamp mode with +10 mV $V_m$ steps (250–500 ms) from −70 mV. AP threshold was measured in $dV/dt$ versus $V_m$ phase plots as the $V_m$ at which AP rise speed reached 10 mV/ms [31,44]. The membrane potential time constant, or apparent tau ("tau"), was measured from the rising phase of 5–10 mV depolarizing steps generated by a 250–500 ms square-pulse positive current steps in a physiological solution without drugs. All parameters were based on at least five traces. Data were acquired with Clampex software (Axon Instruments), digitized at 35–50 kHz, and analyzed offline using pClamp (v10.5; Axon Instruments), Spike2 (v8.1; Cambridge Electronic Design, Milton, UK), OriginPro (v9.5; OriginLab Corporation, Northampton, MA, USA), and SigmaPlot (v14; Grafiti, Palo Alto, CA, USA).

## Drugs

Brain slices were perfused with 100 nM dendrotoxin-K (Cat. No. D4813, Merck/Sigma-Aldrich).

## Tissue fixation and cell visualization

Cells loaded with biocytin during whole-cell patch-clamp recordings were visualized using Alexa 488-conjugated streptavidin (1:1,000) or Cy3-conjugated streptavidin (1:1,000; both Jackson ImmunoResearch, West Grove, PA, USA). Immediately following recordings, slices were fixed in 4% paraformaldehyde and 15% picric acid in 0.1 M phosphate buffer (PB; pH 7.4) at 4 °C for at least 12 h, followed by storage at 4 °C in 0.1 M PB containing 0.05% sodium azide as a preservative. All slices were embedded in 20% gelatin and cut into 50-μm-thick sections using a vibratome (VT1000S, Leica Microsystems, Wetzlar, Germany) in ice-cold PB. Sections were rinsed in 0.1 M PB (3 × 10 min), cryoprotected in 10%–20% sucrose in 0.1 M PB, flash-frozen in liquid nitrogen, and thawed in 0.1 M PB. Slices were then incubated in 0.1 M Tris-buffered saline (TBS; pH 7.4) containing fluorophore-conjugated streptavidin for 2 h 30 min at 22–24 °C, rinsed in 0.1 M PB (3 × 10 min), mounted in Vectashield mounting medium (Vector Laboratories, Burlingame, CA, USA), placed under coverslips, and examined under an epifluorescence microscope (Leica DM 5000 B, Milton Keynes, UK) at ×20–60 objectives.

For single-cell reconstructions, selected sections were incubated overnight at 4°C in a conjugated avidin–biotin horseradish peroxidase solution (1:300; Vector Labs) in TBS (pH 7.4). The enzyme reaction was developed using the glucose oxidase–DAB–nickel method with 3′3-diaminobenzidine tetrahydrochloride (0.05%) and 0.01% $H_2O_2$ as the chromogen and oxidant, respectively. Sections were postfixed in 1% $OsO_4$ in 0.1 M PB, rinsed in distilled water, and stained with 1% uranyl acetate. Dehydration was performed in a graded ethanol series, followed by overnight infiltration with epoxy resin (Durcupan, Merck kft., Budapest, Hungary) and mounting on glass slides. Light microscopic reconstructions were performed with the Neurolucida system (MBF Bioscience, Williston, VT, USA) using a ×100 objective (Olympus BX51, Olympus UPlanFI, Olympus, Tokyo, Japan). Images were collapsed along the z-axis for presentation.

## Immunohistochemistry

Free-floating sections were washed in TBS containing 0.3% Triton-X (TBST) three times over 1 h at 20–24 °C, followed by incubation in a blocking solution containing 20% horse serum in TBST. Antigen retrieval was performed by treating the slices with 1 mg/mL pepsin (catalog #S3002; Dako, Glostrup, Denmark) in 1 M HCl with 0.1 M PB at 37 °C for 5–6 min, followed by rinsing in 0.1 M PB. Sections were then incubated with primary antibodies diluted in TBST for three nights at 4 °C with corresponding fluorochrome-conjugated secondary antibodies (2% blocking serum in TBST) overnight at 4 °C. Between antibody incubations, sections were washed in TBST (3 × 15 min), followed by rinses in 0.1 M PB (2 × 10 min). Slices were then mounted on glass slides with Vectashield mounting medium (Vector Laboratories) for imaging.

The following primary antibodies were used for immunolabeling: goat anti-pvalb (1:1,000; PVG-213, Swant, Burgdorf, Switzerland), mouse anti-spectrin beta-4 (1:100; Invitrogen, Thermo Fisher Scientific, Waltham, MA, USA), rabbit anti-Nav1.6 (SCN8A, 1:300; Alomone Labs, Jerusalem, Israel), rabbit anti-Kv1.1 (KCNA1, 1:100; Alomone Labs), and rabbit anti-Kv1.2 (KCNA2 APC-010, 1:100; Alomone Labs). Prior to confocal microscopy observations, immunolabeling was visualized using the following secondary antibodies: DAM Alexa 488-conjugated donkey anti-mouse (1:400; Jackson ImmunoResearch), DARb Cy3-conjugated donkey anti-rabbit (1:400; Jackson ImmunoResearch), and DaGt A647-conjugated donkey anti-goat (1:200; Thermo Fisher Scientific). Immunoreactivity was evaluated using an epifluorescence microscope with a quasiconfocal option (Zeiss ApoTome 2, Zeiss, Oberkochen, Germany).

## Immunofluorescence image acquisition and analysis

Immunofluorescently labeled sections were imaged using a Leica Stellaris 8 laser-scanning confocal microscope (Leica Microsystems, Biomarker Kft. Budapest, Hungary) with 499-, 554-, and 649-nm lasers to excite the fluorophores, using 504–553-, 559–654-, and 654–750-nm emission filters, respectively.

To quantify immunofluorescence around the soma, 24 radial measurement lines (50 pixels/μm) were placed on the soma such that all lines projected radially from its center into the extracellular space. Immunofluorescence intensity for pvalb (Cy5) and one of the three ion channels (Cy3) was measured along radial lines in single-plane confocal images of the cell soma. The location of the extracellular membrane was defined by the onset of the pvalb signal when moving from the extracellular to the intracellular position along each line. This position was marked as 0 and defined as follows: (average extracellular fluorescence pixel value − average intracellular fluorescence pixel value)/2. For pooled analysis, all lines were aligned at this 0-point.

Ion channel fluorescence from the extracellular space (recorded from the 24 radial lines −3.0 to −0.5 μm relative to the 0-point, in total ca. 3,000 pixels per cell) was averaged and used as a specific reference signal for each cell. This average extracellular value was subtracted from pixel intensities at the membrane (−0.5 to +0.5 μm from the 0-point) and in the intracellular cytoplasm (+0.5 to +1.5 μm from the 0-point). The extracellular site fluorescence (average intensity) did not differ significantly between species (Kv1.1, $n = 31$ cells, $p = 0.06$; Kv1.2, $n = 32$ and 59, $p = 0.835$, Mann–Whitney $U$ test). The intracellular measurement zone was chosen near the extracellular membrane to sample the cytoplasm while avoiding the nucleus, which was not visualized in staining. For axonal analysis, the same extracellular reference value (from 24 lines per cell) was subtracted from fluorescence intensity measurements pixels along the axonal lines.

Z-stack images were acquired from the same cells using a HC PL APO CS2 63×/1.40 oil immersion objective in uni-directional scanning mode. Axonal fluorescence measurement lines were sampled at 50 pixels/μm. AIS immunofluores-cence in z-stack images was analyzed using a custom ImageJ macro (US National Institutes of Health, Bethesda, MD, USA) and a script developed by Dr. Adam Tiszlavicz [86] to determine the AIS's length and its proximal and terminal points (visualized via beta-IV-spectrin immunoreaction). This was achieved by manually marking the points and applying the Euclidean distance formula. Fluorescence intensity was then measured along a polyline connecting these points, capturing the full signal profile of the immunolabeled channels. AIS lengths were normalized by dividing into 10 bins (from proximal to distal). Ion channel immunoreactivity (Cy3) was measured in parallel with pvalb (Cy5) and beta-IV-spectrin (Alexa 488), and axonal fluorescence values were referenced to the average extracellular signal of the corresponding cell, using the same normalization approach applied for somatic radial lines. The custom codes and instructions for using them are available in S1, S2, S3, S4, and S5 Codes.

## Nucleus collection for patch-sequencing

Nuclei were collected from cells exhibiting a fast-spiking phenotype. Prior to data collection, all surfaces, equipment, and materials were thoroughly cleaned using DNA AWAY (Thermo Fisher Scientific) and RNaseZap (Sigma-Aldrich, St. Louis, MO, USA). The perfusion system was flushed with 50 mL of 70% ethanol. Following electrophysiological recording, the pipette was centered on the soma and a small, constant negative pressure was applied to extract the cytosol and draw the nucleus toward the tip. The pipette was slowly retracted under continuous negative pressure while monitoring seal integrity. Once the nucleus appeared at the pipette tip, retraction was accelerated to fully withdraw the pipette from the slice. The pipette containing the internal solution, cytosol, and nucleus was removed from the holder and expelled into a 0.2-mL tight-lock PCR tube (TubeOne, StarLab, Hamburg, Germany) preloaded with 2 μL of lysis buffer comprising 0.8% Triton X-100; (Sigma-Aldrich) and 1 U/μL RNase inhibitor (Takara, Kyoto, Japan). The internal solution contained the following: 110 mM K-gluconate (Sigma-Aldrich), 4 mM KCl solution, RNase-free (Invitrogen), 10 mM HEPES solution (Sigma-Aldrich), 1 mM ATP-Mg (Sigma-Aldrich), 0.3 mM GTP-Na (Sigma-Aldrich), 10 mM sodium phosphocreatine (Sigma-Aldrich), 0.2 mM EGTA (Sigma-Aldrich), 20 μg/mL RNA-grade glycogen solution (Thermo Fisher Scientific), and 0.5 U/μL RNase inhibitor (Takara). The 4-μL sample was briefly spun (2 s), placed on dry ice, and stored at −80 °C until in-tube reverse transcription.

## cDNA amplification and library construction

Libraries were prepared using a protocol based on Smart-seq2 [87,88]. Briefly, cDNA was generated and amplified using 24 PCR cycles for all samples, including negative controls. Indexes and adapter sequences were added to the cDNA,

which was pooled equimolarly and sequenced on an Illumina MiSeq platform (San Diego, CA, USA) in 50-bp single-end read mode. Each sample was sequenced to a depth of ~1–2 million reads.

## Sequencing data processing

Sequencing used 66-bp paired-end reads aligned to the GRCh38 reference genome with annotation file GRCh38_r93 from Ensembl. Alignment was performed using STAR v2.7.11a [89] with the following parameters: --soloType SmartSeq --soloUMIdedup Exact --soloStrand Unstranded --soloFeatures GeneFull. Full gene counts, including introns and exons, produced via STAR were used to compute TPM values for downstream analysis.

## Identification of pvalb neurons based on single-nucleus transcriptomics

Single-nucleus transcriptomic data were analyzed using the Allen Institute's MapMyCells tool (Allen Institute for Brain Science, 2024; RRID:SCR_024672, https://knowledge.brain-map.org/mapmycells/process/). This tool uses gene expression signatures to assign neurons to cortical cell types, enabling identification of pvalb neurons even when *PVALB/pvalb* expression is undetectable (S1 Fig). For human cells, classification used the Seattle Alzheimer's Disease Brain Cell Atlas reference taxonomy with the Deep Generative Mapping algorithm; for mouse cells, the 10× Whole Mouse Brain reference taxonomy was employed with the hierarchical mapping algorithm. Only cells classified as pvalb neurons at $p > 0.9$ (supertype_bootstrapping_probability) were retained for further analysis. Human single-nucleus gene expression profiles showed strong concordance with the Allen Institute's high-resolution reference dataset for pvalb subtypes (subclass correlation coefficient $> 0.5$ using the 10X Whole Human Brain dataset and correlation mapping algorithm; Fig 8).

## Single-cell model

The three-compartment neuron model included cylindrical dendritic and axonal compartments and followed a previously established formalism [28]. The model comprised a soma; a 140-µm-long, 0.8-µm-diameter axon; and a 200-µm-long, 1-µm-diameter dendrite. Axonal cable parameters were as follows: axon Ra = 0.8–1.1 MΩ·µm; axon Cm = 6 fF/µm². Dendritic Cm was set to 12 fF/µm², i.e., 2-fold higher than that of the axon. The somatic compartment had a resistance of 700 MΩ and a Cm of 25 pF. The leakage reversal potential was set to −68 mV. Rheobase and voltage sag parameters were chosen to match the basic response properties and carefully adjusted using the experimental data to match the simulated and recorded voltage traces. All intrinsic voltage-dependent currents were calculated using the following equation:

$$I_i = g_i m_i^p h_i (E_i - V),$$

where $i$ represents individual current type; $g_i$ is the maximal conductance of the current; $m_i$ and $h_i$ are the activation and inactivation (either first-order or absent) variables, respectively; $p$ is the activation exponent; and $E_i$ is the reversal potential. Activation ($m$) and inactivation ($h$) kinetics were modeled based on the following equation:

$$\frac{dx}{dt} = \frac{x_\infty(V) - x}{\tau_x(V)},$$

where $x$ represents $m$ or $h$. The voltage-dependent steady-state activation and inactivation are described by the following sigmoid function:

$$x_\infty(V) = \frac{1}{2} + \frac{1}{2}\tanh\left(\frac{V - V_{x,1/2}}{V_{x,sl}}\right).$$

The midpoint $V_{x,1/2}$ and slope $V_{x,sl}$ parameters of the sigmoids and other kinetic parameters were presented in previous study [63]. The apparent tau ($\tau$) values of activation and inactivation were modeled as bell-shaped functions of the $V_m$ ($V$) according to the following equation:

$$\tau_x(V) = (\tau_{x,max} - \tau_{x,min})\left[1 - \tanh\left(\frac{V - V_{\tau x,1/2}}{V_{\tau x,sl}}\right)^2\right] + \tau_{x,min}.$$

EPSP waveforms were generated by EPSCs with a time to peak of 0.5 ms, a decay time constant of 3 ms, a reversal potential of 0 mV, and peak conductances of 6–11 nS.

## Statistical analysis

Results are presented as medians and IQRs. Statistical comparisons were performed using the Student $t$ test, the Mann–Whitney $U$ test, or Wilcoxon signed-rank test. For multiple comparison ANOVA on ranks (Kruskal–Wallis $H$ test) with Dunn's *post hoc* pairwise test, or PERMANOVA and the Mann–Whitney $U$ test with Bonferroni correction was used. A mixed-design ANOVA with a post hoc pairwise test using Holm's correction for repeated measures was used to test for differences within and between species, as well as the interaction between them (i.e., whether changes in measurement values over time differ significantly between human and mouse species). Proportions were compared using the chi-squared test with Benjamini–Hochberg adjustment at a 5% significance level. Correlations were assessed using Spearman's rank-order method.

## Copyright and clip art

The clip art images (the rodent and human silhouette insets) were downloaded from https://commons.wikimedia.org. These images were modified in terms of color and relative dimensions to create the final figures. The other clip art images (the schematics) were drawn by the authors.

## Supporting information

**S1 Fig. Model cell simulation incorporating AIS parameter variability, showing no difference between species.**
**(A)** Increasing AIS Nav1.6 conductance (gNav1.6, 15–40 nS/µm²; abscissa) progressively lowered the AP firing threshold across multiple Kv1 activity levels (0–10 nS/µm²). AIS length = 20 µm; axon Ra = 0.8–1.1 MΩ·µm; axon diameter = 0.8 µm. **(B)** Proximal–distal positioning of the AIS regulated the AP firing threshold within the range observed anatomically. Simulations used square-pulse current steps to mimic $V_m$ changes (as shown in Fig 8). **(B1)** $V_m$ traces from simulations in the model cell with the AIS positioned either directly at the axon hillock (0 µm; purple) or 10 µm distally (brown), with consistent AIS length (20 µm), axon Ra (0.8–1.1 MΩ·µm), axon diameter (0.8 µm), and channel conductances (20 nS/µm² for Nav1.6 and Kv1). **(B2)** Firing threshold values plotted for AIS locations and across three Ra conditions (0.8–1.1 MΩ·µm, indicated by symbols). A more distal AIS position consistently resulted in a lower AP firing threshold, replicating the real-neuron correlation observed in Fig 3C₁. **(C)** Heatmap shows the relationship between axon longitudinal conductance (normally reduced by axon diameter increase) and axon diameter simulating a proportional (1.3-fold) increase in both Kv1 and Nav1.6 ion conductance with increasing axon diameter, uncoupling parameters that are typically correlated in real neurons. Increasing only the Ra lowered the AP firing threshold. Conversely, increasing the ion channel conductance with an increased AIS membrane area, while keeping the channel density fixed at 20 nS/µm² for Nav1.6 and Kv1, increased the firing threshold in the model cell. This indicates the strong effect of Kv1 channels on the threshold.
(TIF)

**S1 Table. List of neocortical pvalb cells included in the study.**
(DOCX)

**S1 Data. Source data for** Figs 1–4**.**
(XLSX)

**S2 Data. Source data for** Figs 5–7**.**
(XLSX)

**S3 Data. Source data for** Fig 8**.**
(XLS)

**S4 Data. Source data for** Fig 9 **and** Table 2**.**
(XLS)

**S1 Code. Illustrated instructions for running the ImageJ and excel analysis code.**
(PDF)

**S2 Code. Custom ImageJ macro for AIS analysis.**
(IJM)

**S3 Code. Custom ImageJ macro for radial line analysis.**
(IJM)

**S4 Code. Excel script for AIS analysis.**
(XLS)

**S5 Code. Excel script for radial line analysis.**
(BAS)

## Acknowledgments

We thank both Dr. Ferhan Ayaydin at the Functional Cell Biology and Immunology Advanced Core Facility, Hungarian Center of Excellence for Molecular Medicine, and Szeged University and Dr. Fanni Nedenyi in the HCEMM Scientific Computing core facility Modelling and statistics Unit for help with statistical analyses. The authors would like to thank Enago (www.enago.com) for the English language review.

## Author contributions

**Conceptualization:** Viktor Szegedi, Karri Lamsa.

**Data curation:** Ádám Tiszlavicz, Viktor Szegedi, Daphne K. Welter, Jonathan M. Landry, Vladimir Benes, Karri Lamsa.

**Formal analysis:** Emoke Bakos, Ádám Tiszlavicz, Viktor Szegedi, Abdennour Douida, Szabina Furdan, Daphne K. Welter, Jonathan M. Landry, Attila Szucs.

**Funding acquisition:** Attila Szucs, Karri Lamsa.

**Investigation:** Emoke Bakos, Ádám Tiszlavicz, Viktor Szegedi, Abdennour Douida, Szabina Furdan, Attila Szucs.

**Methodology:** Emoke Bakos, Ádám Tiszlavicz, Daphne K. Welter, Balazs Bende, Gabor Hutoczki, Pal Barzo, Vladimir Benes, Attila Szucs.

**Project administration:** Viktor Szegedi, Balazs Bende, Karri Lamsa.

**Resources:** Abdennour Douida, Gabor Hutoczki, Pal Barzo, Gabor Tamas.

**Software:** Attila Szucs.

**Supervision:** Viktor Szegedi, Karri Lamsa.

**Validation:** Emoke Bakos, Viktor Szegedi, Abdennour Douida, Vladimir Benes, Attila Szucs, Karri Lamsa.

**Visualization:** Emoke Bakos, Viktor Szegedi, Abdennour Douida.

**Writing – original draft:** Karri Lamsa.

**Writing – review & editing:** Karri Lamsa.

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
