## [Editor Report · Decision Letter 0]

13 Nov 2024

Dear Dr Lamsa,

Thank you for submitting your manuscript entitled "Specialized axon initial segment enables low firing threshold with rapid action potential output in fast-spiking interneurons in the human neocortex" for consideration as a Research Article by PLOS Biology.

Your manuscript has now been evaluated by the PLOS Biology editorial staff, as well as by an academic editor with relevant expertise, and I am writing to let you know that we would like to send your submission out for external peer review.

Once your full submission is complete, your paper will undergo a series of checks in preparation for peer review. After your manuscript has passed the checks it will be sent out for review. To provide the metadata for your submission, please Login to Editorial Manager (https://www.editorialmanager.com/pbiology) within two working days, i.e. by Nov 15 2024 11:59PM.

Kind regards,

Taylor

Taylor Hart, PhD,

Associate Editor

PLOS Biology

thart@plos.org

---

## [Decision Letter · Decision Letter 1]

9 Jan 2025

Dear Dr Lamsa,

Thank you for your patience while your manuscript "Specialized axon initial segment enables low firing threshold with rapid action potential output in fast-spiking interneurons in the human neocortex" was peer-reviewed at PLOS Biology. Your manuscript has been evaluated by the PLOS Biology editors, an Academic Editor with relevant expertise, and by several independent reviewers.

As you will see in the reviewer reports, which can be found at the end of this email, although the reviewers find the work potentially interesting, they have also raised a substantial number of important concerns. Based on their specific comments and following discussion with the Academic Editor, it is clear that a substantial amount of work would be required to meet the criteria for publication in PLOS Biology. However, given our and the reviewer interest in your study, we would be open to inviting a comprehensive revision of the study that thoroughly addresses all the reviewers' comments. Given the extent of revision that would be needed, we cannot make a decision about publication until we have seen the revised manuscript and your response to the reviewers' comments. Your revised manuscript would need to be seen by the reviewers again, but please note that we would not engage them unless their main concerns have been addressed.

As you will see, the reviewers think that the study is well-executed technically and provides important insights. However, the reviewers were concerned over the reliance on antibody binding immunofluorescence for quantification and mentioned the need to consider additional types of channels in the analysis. R2 and R3 raise several significant concerns over the level of support for the study's major findings, especially limitations in the reported electrophysiology data and inconsistencies between the data and model. We think that substantial new experimental data will likely be required to address these concerns, in addition to textual changes to clarify all concerns raised by the reviewers.

We think you should include electrophysiology experiments performed during pharmacological blocking of Kv1 channels, as well as using small hyperpolarizing steps as suggested by R3. In addition, you should make textual changes to clarify all concerns raised by the reviewers, especially the role of other channel types and differences between the experiment and simulations.

We appreciate that these requests represent a great deal of extra work, and we are willing to relax our standard revision time to allow you 6 months to revise your study. Please email us (plosbiology@plos.org) if you have any questions or concerns, or envision needing a (short) extension.

**IMPORTANT - SUBMITTING YOUR REVISION**

*Resubmission Checklist*

*Published Peer Review*

*PLOS Data Policy*

*Blot and Gel Data Policy*

Sincerely,

Taylor

Taylor Hart, PhD,

Associate Editor

PLOS Biology

thart@plos.org

REVIEWS:

Reviewer #1: This is an excellent study by Bakos and colleagues that provide a detail and thorough analysis of spiking properties of human and mouse interneurons, providing a potential explanation for how human cells maintain fast firing rates despite slower a membrane time constant. The authors effectively demonstrate that the AIS is longer in human cells compared to mouse, but has similar diameters. AP properties are examined thoroughly with very rigorous electrophysiology, showing that AP threshold is lower in human cells. This correlates with a reduction in Kv1 channels in the AIS of human Pv+ cells.

The work is well written and illustrated. I have only a few very minor suggestions that should be addressable with revisions to text or revisions to already existing data/datasets:

1) The authors should consider expanding the limitations section to note explicitly that immunostaining relies on antibodies that can only be validated by knockout in non-human species. Alternatively or additionally, an analysis of sequence conservation of mouse vs human Kv1 channels at the antibody binding site would help validate the lack of staining. While it is clear that Pv+ cells largely lack Kv1 labeling in human tissue, there is a possibility that the positively marked initial segment staining is nonspecific. This appears to be more of a concern with Kv1.2 as it appears that there is sub-membranous staining in somata with this antibody.

2) Much of the staining display may be over-saturated, as one cannot discern typical "railroad" patterns for membrane-bound proteins in the AIS. This is always difficult with such small structures, and may be a display issue, but it also relates to my concern about binding specificity. If images could be checked for saturation and corrected if present, that would be very useful.

3) From examples, it appears that human cells have an AP initiation site that is more biophysically separated from the soma in human compared to mouse. This is inferred from the AP phase plane examples in Fig 1 B2 and C2. In human there is an inflection point in the rising phase of the AP that is usually associated with more distal AP initiation. An analysis of the prevalence of these sorts of dynamics could be useful, for example by examining the AIS and somatic components of the rising phase with analysis of dV^2/dt.

4) Just recently, additional leak/stretch sensitive potassium channels have been identified at the AIS in mouse (TREK, TRAAK). Furthermore there are Kv7 channels that are very important for resting Vm and threshold in the AIS. These may be important for subthreshold aspects of excitability in the AIS, perhaps more than Kv1. I fully appreciate that patch seq is both difficult and limited in its interpretation, but if the authors have the data already, an expansion of the display to include all potential potassium-fluxing players in the AIS would be very beneficial.

Reviewer #2: In the study conducted by Bakos and colleagues, human parvalbumin-expressing (PV) interneurons from various neocortical regions were compared to mouse PV interneurons. The primary finding indicates that human interneurons exhibit a more hyperpolarized voltage threshold for action potential generation. Geometrical analysis of the axon initial segment, combined with Na+ and K+ channel protein staining, revealed a significantly lower expression of Kv1.1 in humans. Further analysis using a public database and patch-sequencing of the recorded interneurons showed that KCNA1 gene transcripts are significantly lower in humans. The authors continued to develop a computational model to replicate some of the experimental data. The recordings are technically excellent, and the authors gathered an impressive amount of data on parvalbumin interneuron properties in the human neocortex, employing a diverse range of techniques from electrophysiology to single-cell RNA sequencing, often from the same single interneurons from which functional data were collected. However, this reviewer's enthusiasm for the paper is tempered by several factors: the findings are largely descriptive and correlative. The claim that low Kv1 expression in the human axon initial segment accounts for the lower threshold requires additional experiments, and the role of Kv1 and the axon initial segment needs further substantiation.

1. Quantifying voltage threshold alone (p. 3) is insufficient to understand the electrophysiological differences between human and mouse interneurons. Kv1 is a fast activating voltage-gated potassium channel known for its role in rapid repolarization and is also slowly inactivating, as described in numerous landmark studies (10.1016/j.neuron.2008.03.003; 10.1016/j.neuron.2012.12.020). Relevant parameters missing include, for example, the delay from current injection onset, firing rates at current threshold (rheobase), current thresholds, action potential amplitude, action potential half-width, afterhyperpolarization, firing adaptation, frequency-dependent width adaptation, etc. Importantly, if the initial segment excitability is the dominant driver in the threshold the components of the action potential phase plane need to be quantified. An integrated view on the physiology is required.

2. If the lack of Kv1 at the AIS indeed accounts for a more hyperpolarized voltage threshold in human neurons, pharmacological block should abolish the species-dependent difference. At a minimum, the authors should apply dendrotoxin (Dtx-I, a specific Kv1 channel blocker) to assess its contribution to the action potential parameters (and quantify more than threshold, see #1). Since Kv1 is expressed at both the soma and axon membrane, unlike claims made here in the manuscript, the channel blocker could be applied locally to the initial segment to segregate axonal from somatic Kv1 contribution to membrane excitability. The approach furthermore may show whether other identified differences (e.g. AIS length) contribute (see #3)

3. Whether the ~3 µm longer axon initial segment plays a role in the action potential threshold is unclear. The correlation analysis is not in line and voltage threshold instead scales with the start distance from the soma (lines 223-235). Computational models deliver only a corollary and cannot be interpreted as conclusive evidence (see point #2). A major concern is that the parameter range is inconsistent with the data; a 10 µm length increase in the model reduces the voltage threshold at the soma by 3 mV (compared to the experimental results putatively showing a ~4 µm more distally positioned initial segment length correlates with no ~10 mV lower threshold). Last but not least, the simulations predict large changes in AP amplitude with lengthening, which are not seen (or not reported) in the data.

4. There are major concerns about the language used for the electrophysiology. A few examples, 'firing threshold' is a vague term. Voltage- and current thresholds are absolute measures but depend on many technical confounding factors including membrane potentials and duration of the current injections. Furthermore, voltage-gated channels are not 'inhibitory' (e.g. line 59 and elsewhere) since that is used to describe the impact of synaptic conductance. Also "Kv activity" is unclear and ambiguous (line 381). Are the authors referring to channel opening, or perhaps channel conductance densities? Other issues are noted at line 98 (and throughout the entire manuscript) where the authors write "electrical resistance". Neurons have an input resistance, reflecting the local cell's membrane response upon current injections, encompassing aspects of geometry as well as the intracellular and membrane resistances, in units of Ohm. Membrane resistance, however, is a unity measure, referring to the specific feature of the membrane (Ohm cm2). At p. 9 they write about "real-life pvalb neurons", departing from scientific language. Please rigorously edit the manuscript an expert to make the accessible to expert and broader audience.

5. The strongest correlation was found between the membrane time constant and the voltage threshold (line 180). Lines 492-500 are meant to provide an explanation, but incomprehensible and extremely speculative without having data on the morphology or the passive membrane properties (all voltage-gated conductances being blocked). Time constant also depends on the channel composition. The unexplained time constant requires pharmacological approaches

Reviewer #3: Bakos et al. report the results of an interesting study assessing differences between parvalbumin-positive (PV) neuron in slices from human and mouse neocortex. Given that data from human PV cells are very scarce, the comparison done in this paper could be very useful for many colleagues in the neuroscience community. There are several points that the authors should carefully address before the study can be acceptable for publication. Please, see my comments below.

Major

1) The study compares AP threshold across species using the definition: "the Vm at which the AP initial rise speed reaches 10 mV/ms (Fig. 1A1)." However, the differences between mouse and human cells could vary depending on how AP threshold is defined. Since these differences are central to the paper's conclusions, the authors should clarify why this specific definition was chosen. Would the human-mouse differences change if the threshold were defined using 20 or 5 mV/ms instead of 10 mV/ms?

2) As shown in Fig. 2B, the authors measured the membrane time constant using depolarizing current steps. This approach is problematic because depolarizing steps often activate subthreshold voltage-gated channels (below the AP threshold). Since the membrane time constant is a passive property, it should be measured under conditions that avoid activating voltage-gated channels. Typically, this is achieved using small hyperpolarizing current steps, which minimize such activation.

Given that the authors have measured multiple membrane properties, such as input resistance and capacitance, their dataset likely includes responses to hyperpolarizing current steps, which are essential for accurately estimating the membrane time constant in the hyperpolarizing range. This information should be incorporated into the manuscript to ensure a proper and "uncontaminated" analysis of the time constant. Otherwise, the between-species comparison may be confounded by using a method that measures time constants affected by voltage-gated channel activation.

Indeed, Fig. 2B1 suggests that the depolarizing steps in human neurons (but not in mice) produce a voltage "hump," indicative of voltage-gated channel activation. Hyperpolarizing steps could of course activate hyperpolarization-activated currents (Ih) mediated by HCN channels, which are widely expressed and have been reported in human PV neurons in published studies. However, HCN is sometimes considered part of the leak channels that regulate input resistance (Rin) and tau near resting membrane potentials, hence HCN channel activation is expected when measuring these passive properties. In contrast, depolarization activates a wide array of voltage-gated channels, which can significantly skew measures of Rin and tau.

3) Lines 174-177: Regarding point 2 above, the lack of correlation between Rin and tau in human neurons, as well as the similar tau values in cells with high and low Rin, is unexpected. Tau is typically proportional to the product of membrane resistance and capacitance (Tau = Rm * Cm), so tau and Rin are generally correlated, with higher Rin associated with higher tau values.

One possible explanation for this discrepancy is that the membrane time constant was estimated using depolarizing steps. As shown in Fig. 2B1, the depolarizing steps in human neurons appear to be "contaminated" by the activation of voltage-gated channels, which could disrupt the expected correlation between these passive properties. To resolve this, it is essential to measure Rin and tau using small hyperpolarizing current steps, minimizing voltage-gated channel activation. This would confirm whether the observed human-mouse differences are due to genuine differences in passive membrane properties or the differential activation of voltage-gated channels.

4) The experiments examining AIS properties (Fig. 3) are very elegant, and I commend the authors for this impressive work. However, I am somewhat unclear about whether the authors propose that the actual AP initiation site is located specifically within this anatomically defined axonal compartment (as identified by beta-spectrin labeling) and how this relates to the differences in AP threshold.

In particular, the interpretation of the correlations in Fig. 3C is not entirely clear. The strongest correlation is between AP threshold (APthr) and AISa (p = 0.012). Does this suggest that AP initiation occurs in this segment of the axon? By contrast, the correlation with AISb—the AIS segment supposedly well-defined by beta-spectrin labeling—is not significant. This discrepancy raises questions about the precise role of these AIS regions in AP initiation.

I believe the interpretation of these results requires more careful consideration in the manuscript. Additionally, the implications for the spatial location of the AP initiation site should be discussed in greater depth. It would be valuable to reference and compare these findings with published studies that have directly recorded from the axon to assess the AP initiation site, including cases where axonal AP thresholds were measured directly, even if not in PV cells from the human cortex. This would provide a broader context and strengthen the discussion.

5) The rationale for investigating the relationship between AP threshold and axon diameter or Nav1.6 distribution needs clarification. While axon diameter is important for subthreshold voltage propagation, with active membrane propertiesthe situation could be quite different. The implications of Nav1.6 distribution patterns or gradients along the AIS (proximal to distal) should also be better explained. It is unclear how the reported AP threshold differences between human and mouse neurons relate to specific Nav1.6 density gradients or distributions, and clarifying these connections would help. Lastly, species differences in AP thresholds might reflect variations in Nav1.6 density, but immunohistochemistry has limitations for comparing channel densities across species due to antibody variability and labeling efficiency. Acknowledging these issues and suggesting alternative methods would strengthen the discussion.

6) The immunohistochemistry/mRNA data for Kv1 channels suggest that human PV neurons have very low levels of Kv1.1 in the AIS membrane. However, the quantification method may exaggerate species differences. In Fig. 6A and B, data are normalized to pre-AIS signal levels, assuming low beta-spectrin levels pre- and post-AIS. It's unclear if Kv1 channel density truly differs across these compartments, making species comparisons with this normalization approach challenging.

For example, it's hard to conclude that Kv1.2 levels in the AIS are low; Fig. 6B5 mainly shows that these levels remain constant relative to pre-AIS, with an F/Fpre-AIS ratio near 1. Both Fig. 6A and B suggest Kv1 levels are low relative to beta-spectrin in the AIS of both species, but more so in humans. By the same reasoning, one might infer that PV levels are lowest in the AIS. However, Fig. 6 does not provide enough evidence to determine if PV levels in the AIS are low compared to other AIS proteins or across species. The patch-seq data might clarify some of these uncertainties, though I have reservations, as discussed in point 7.

7) Including patch-seq data, as in Fig. 7, could provide highly valuable insights, but the presentation of the data raises some questions. For instance, if the blue color in the heatmap represents TMP values close to zero, it suggests that nearly half of the neurons recorded in human and mouse cortical slices lack detectable PV mRNA. Were the cells included in the patch-seq experiments confirmed to have PV protein immunoreactivity? This could be challenging, as extracting the nucleus for patch-seq might disrupt the neuronal soma, complicating immunodetection of PV protein.

If all cells in the Fig. 7A1 heatmap were confirmed PV protein-positive, the absence of detectable PV mRNA in ~half of these neurons would indicate that PV mRNA is not uniformly detected in PV neurons from human and mouse cortices. This raises concerns about interpreting patch-seq results, as the absence of PV mRNA—the primary marker for this cell type—suggests that RNA expression profiles should be analyzed with added caution. Similarly, SLC6A1, which encodes the vesicular GABA transporter GAT1, a marker for all interneuron subtypes, is detected in only ~half of human neurons. This further complicates the interpretation of patch-seq data.

Additionally, the statement, "mRNA analysis of patch-sequenced human and mouse pvalb neurons (upper schematic illustrates the protocol) revealed the absence of KCNA1 expression in all human neurons, whereas it was robustly expressed in many mouse cells," seems inaccurate. According to the heatmap, KCNA1 mRNA is undetectable in 10 of the 16 mouse PV cells, with most other cells showing only low expression levels. This discrepancy should be addressed to ensure accurate representation of the data.

8) While I greatly value computational modeling data, in this study, it would be crucial to demonstrate that at least some of the findings from the model are predictive of experimental differences in spiking activity and firing patterns between PV cells in mouse and human cortices. Including experimental data comparing spike trains would significantly enhance the manuscript, as some readers may perceive the scope as too narrow.

Specifically, the observed mouse-human differences in AP properties likely have important consequences for input-output transformations in neurons. This could be addressed in model neurons by varying the amount and temporal patterns of excitatory current. Additionally, the authors could conduct current-clamp experiments in slices to test the model's predictions regarding firing pattern differences between mouse and human PV cells.

Minor

- Lines 146-156: the authors compare AP threshold between human frontal and temporal cortex slices, but it is not clear if AP threshold differs in comparison with any other areas, nor what are was recorded in slices from mouse cortex. Please clarify.

- Lines 137-138, please state what area of mouse neocortex was studied.

- Lines 140-141: "to identify the lowest threshold that triggered APs"

This statement sounds odd. It appears that a cell can have multiple AP thresholds and the authors somehow to pick the lowest of all. Cell could have different thresholds to trigger an AP, but if so, this happens in different membrane compartments, like soma versus mid-dendrite versus distal dendrite. Since the experiments in this study do not involve recordings from different compartments, the sentence is not clear…

- Lines 155-156: "The results showed that human pvalb neurons have a lower AP firing threshold potential in response to excitation in the soma than mouse pvalb neurons."

While this is true, the authors should explain under what type of excitation the differences were observed (current steps/pulses?).

- Lines 159-161:

The text would be clearer if the cortical area is indicated.

- Lines 162-164: "We measured the average Rin and Cm in neurons in Vm steps of +5 to +10 mV (elicited by square pulse currents), which were excitatory but below the AP firing threshold (subthreshold) when applied at −70 mV (Fig. 2A1)."

It is more appropriate to say that the current pulses were depolarizing, since depolarizing steps that are subthreshold can have a mix of excitatory and inhibitory effects.

- Lines 210-213: "from 28 human and 22 mouse cells recorded in the whole-cell clamp, the intact soma and full-length AIS were successfully recovered (demonstrated by beta-IV-spectrin immunofluorescence) in 13 human and 14 mouse neurons."

I may be misunderstanding this sentence, but the numbers of cells with recovered axon appear low, given that, typically, when FS interneurons are filled with biocytin recovery of large fractions of axon is common (usually all the axon that "survived" slicing). Hence, finding axonal labeling beyond the AIS should be relatively common and achieved in many recorded cells, even if the full complement of axon branches is not recovered. I am wondering if this sentence actually refers to cells in which beta-IV-spectrin immunofluorescence was successful? If recovery of the axon is much lower in FS cells recorded in human cortex slices, compared to mice, then this should be clearly stated.

-Fig 1A2-3: is it not clear if the analysis of PV+ axon segments in A3 was done in the reconstructed PV neuron shown in A2

---

## [Decision Letter · Decision Letter 2]

29 Aug 2025

Dear Dr Furdan,

Thank you for your patience while we considered your revised manuscript "Specialized axon initial segment enables low firing threshold and rapid action potential output in fast-spiking interneurons of the human neocortex" for publication as a Research Article at PLOS Biology. Your revised study has been evaluated by the PLOS Biology editors, the Academic Editor and several of the original reviewers.

In light of the reviews, which you will find at the end of this email, we would like to invite you to revise the work in response to the reviewers' reports. However, want to emphasize that some of the remaining issues are substantial and these reports require a thorough response.

While the reviewers commended the effort put into the revisions, Reviewer 2 said that substantial concerns remain about the strength of evidence for the major claims. We think that to be a stronger candidate for publication at PLOS Biology, the study would further revisions to address the issues highlighted in this round of review, especially Reviewer 2's Major Point 3.

Given the extent of revision needed, we cannot make a decision about publication until we have seen the revised manuscript and your response to the reviewers' comments. Your revised manuscript is likely to be sent for further evaluation by all or a subset of the reviewers.

**IMPORTANT - SUBMITTING YOUR REVISION**

*Re-submission Checklist*

*Published Peer Review*

*PLOS Data Policy*

*Blot and Gel Data Policy*

Sincerely,

Taylor

Taylor Hart, PhD,

Associate Editor

PLOS Biology

thart@plos.org

REVIEWS:

Reviewer #2: In this revised manuscript, the authors have undertaken several new experiments, which is commendable and has the potential to enhance the overall quality of the work. However, in addressing the concerns raised by this reviewer, the rebuttal and the interpretation of the new data appear to be inconsistent with the findings. My principal concern remains that the manuscript does not provide direct evidence to support its central claim—that the properties of the "specialized axon initial segment" (AIS) account for the observed threshold differences.

- Major Point 1: The authors have now quantified a greater number of parameters, as presented in the updated Table 1. Nevertheless, neither the rebuttal nor the manuscript offers a clear interpretation or contextualization of these data. For instance, what is the significance of the observed increase in half-width duration? Similarly, how should the differences in latency be understood?

- Major Point 2: The original concern pertained to whether differences in Kv1 expression at the AIS are sufficient to explain the electrophysiological properties observed. The authors have conducted new experiments involving the selective Kv1.1 blocker dendrotoxin-k (DTX-k), as shown in the new Table 2 and Figure 9. While these experiments are valuable, the rebuttal fails to articulate the insights gained from them and draws erroneous conclusions. A key issue lies in the statistical analysis of the data presented in Table 2 and Figure 9. The authors compare DTX-k versus baseline within each species and then statistically assess "differences of differences" in Figure 9, which is methodologically flawed. To properly evaluate whether Kv1 differential expression accounts for the distinct threshold and other action potential parameters, a two-way repeated measures analysis of variance is required. Subsequent post-hoc comparisons of the absolute data should then be used to determine whether the means of the action potential parameters in the presence of the blocker are comparable between human and mouse neurons.

- Major Point 3: The authors assert that "given that the AIS is longer in humans than in mice, we still argue that it likely contributes to the threshold difference." However, no evidence is provided to substantiate the claim that AIS length directly influences voltage threshold. In fact, the data in Table 2 suggest that, in the presence of DTX-k, action potential thresholds are similar across species, despite the longer AIS in humans. This implies that the modest length difference of approximately 3 µm does not contribute to a lower threshold. This conclusion is further supported by the authors' own immunofluorescence analysis (Figure 6), which demonstrates that Kv1 channels are not exclusively localized to the AIS membrane but there is also a lower fluorescence detectable at the somatic membrane. Indeed, a global reduction in KCNA1 expression and Kv1 protein levels would also affect somatic membrane properties, potentially explaining the strong correlations observed with somatic rise times.

- Major Point 3, Continued: The authors acknowledge that the neuronal model lacks sufficient detail and have initiated a new collaboration with Dr Giugliano to address this. However, it is unclear how a future study would resolve the fundamental concerns raised in the current manuscript, particularly given the aforementioned arguments that AIS length differences alone do not play a significant role.

Reviewer #3: I thank the authors for their detailed response to my previous comments. I think the revised manuscript will be significantly improved. Before the manuscript is ready for publication, there are still a few points that need to be addressed. Please, see below.

1) In their responses to my first comment: "We used depolarizing voltage steps for tau measurement for two reasons: …" the authors justify why depolarizing steps were used, characterizing special conditions inconsistent with the "traditional" meaning of Tau, the membrane time constant of a neuron. While the authors' approach could be, in principle, OK, they need to clearly specify in the text of the manuscript (to avoid confusing future readers, especially non-expert readers of PLoS Biology) that the meaning of tau in their manuscript is different from the consensus tau. While the functional meaning of the time constant of a neuron is partly ambiguous (see for instance, A brief history of time (constants), C Koch, M Rapp, I Segev 1996, PMID: 8670642), the authors should clarify that their measurement named as "tau" diverges from the meaning of the membrane time constant "tau" measured under standard conditions (see, for instance, Spruston and Johnston 1993, PMID: 1578242).

The membrane property typically named "tau" relates to the passive membrane property. While tau is obviously not a purely passive property in any real neuron, is quite different from the measurement that the authors name "tau" in this manuscript. Simply changing the name used in the manuscript and specifying why the conditions of the measurements do not assess the standard "tau" would be sufficient.

2) The responses to my second comment: "…Again, we emphasize that our study focuses on membrane properties between the resting potential and AP firing threshold …", still suggest the risk of confusing future readers. Indeed, the trajectory of the membrane potential between resting and the threshold voltage to trigger a spike is a very important and interesting property to study and to compare between PV interneurons across species. But this does not mean that the time constant measured by the authors could be named "tau" without stating some qualifier upfront. As remarked in my previous comment, the authors need to make revisions to make clear to future readers that the time constant they have measured most likely does not reflect "tau" in the standard/consensus meaning of the word.

3) The authors response: "moderate sample sizes are insufficient to draw strong statistical conclusions about parameter associations" seems reasonable, but the revised text of the manuscript is weak/vague in this regard. It should be revised to make sure the future readers clearly understand which conclusions are not possible to make from the authors' data, because of the small sample size and large cell-to-cell variability.

4) I don't think the authors properly responded my question of why they propose that axon diameter is crucially relevant for active propagation of signals (mediated in part by Nav1.6 channels) … Their response states some immunohistochemistry findings related to the subcellular distribution of Nav1.6 immunofluorescence with the AIS … : "We performed new study quantifying AIS Nav1.6 immunofluorescence, as noted in our response to point 4. The new analysis (revised Figure 5) tested whether Nav1.6 signal intensity peaks at the first or second half of the AIS proximal-distal axis…" But how and why does this matter in terms of role of diameter in signal propagation through the AIS? Please clarify the sentence/s.

5) The complexity in interpreting the patch-seq data (PVALB and SLC6A1 expression levels, among others) highlights the possibility that the quality of the patch-seq data was not sufficient to properly support the claims made by the authors. This should be more strongly emphasized in the revised manuscript.

6) The authors state: "although our mRNA analysis, performed and supervised by the EMBL GeneCore facility and Dr. Vladimir Benes, reveals transcribed gene levels, the protocol we currently use may be less sensitive than the Allen Institute's protocol for detecting genes expressed". While this situation is understandable, the statement highlights the limitations in comparing/aligning the patch-seq data from the present study with the Allen Institute databases. This limitation should be clearly mentioned in the revised manuscript, ideally including a brief technical explanation of why the protocol employed in their study is less sensitive.

7) Regarding my minor comments about recovered AIS beta-IV-spectrin immunoreactivity etc., the statements made by the authors in their responses should be incorporated in the text of the manuscript, either in methods, in results or both sections, depending on the specific point.

---

## [Decision Letter · Decision Letter 3]

12 Nov 2025

Dear Dr Lamsa,

Thank you for your patience while we considered your revised manuscript "Specialized axon initial segment enables low firing threshold and rapid action potential output in fast-spiking interneurons of the human neocortex" for publication as a Research Article at PLOS Biology. This revised version of your manuscript has been evaluated by the PLOS Biology editors, the Academic Editor and one of the original reviewers.

Based on the review, we are likely to accept this manuscript for publication, provided you satisfactorily address the remaining point raised by the reviewer. Please also make sure to address the following data and other policy-related requests.

IMPORTANT: Please ensure that your next revision addresses the following editorial requirements:

---------------

**Title:

-- We would like to suggest a tweak of your title to streamline it and emphasize the main findings. Does this alternative formulation work for you?

"Adaptations of the axon initial segment in fast-spiking interneurons of the human neocortex support low action potential thresholds"

**Financial disclosure statement:

-- Please add links to the funding agencies in the Financial Disclosure statement in the manuscript details.

**Ethics:

-- Thank you for including information about your study's ethical approvals. As your ethics statement does not explicitly mention the mouse work, please confirm if these approvals covered both the human and animal work, and add the necessary information if not.

-- For animal work, we require the full name of the IACUC/ethics committee that reviewed and approved the animal care and use protocol/permit/project license. Please also include an approval number, and the specific national or international regulations/guidelines to which your animal care and use protocol adhered. Please note that institutional or accreditation organization guidelines (such as AAALAC) do not meet this requirement.

-- Please make the Ethics Statement a separate, independent (and the first) subheading in the Materials & Methods section. Please separate the license information into a different sub-heading.

**Data and Data Availability:

-- We noticed that your data availability statement in the submission form and in the text do not precisely match (in the form, you say that all data are available without restriction, but in the text, you say that underlying data and code are available upon request). While we appreciate that you have included the data underlying the figures in the supplement, we strongly encourage you to make all publicly available and to bring these statements into alignment.

-- Again, thank you for providing the data underlying the figures in the supplement. However, we would like to request that you re-name these files to match our standard format. Please make sure that all data files are uploaded as 'Supporting Information' and are referred to invariably using the following format verbatim: S1 Data, S2 Data, etc. in the file names and descriptions throughout your manuscript. For example, you can rename the file currently titled "Metadata Figs 1-4 New.xlsx" as "S1_Data.xlsx".

-- Related to this, please add descriptions of the supplemental data files to the manuscript, and please include in the figure legends where the underlying data can be found, e.g. “The data underlying this Figure can be found in S1 Data”

**Code availability:

---------------

We expect to receive your revised manuscript within two weeks.

*Published Peer Review History*

*Press*

Sincerely,

Taylor

Taylor Hart, PhD,

Associate Editor

thart@plos.org

PLOS Biology

Reviewer remarks:

Reviewer #2: I am satisfied the authors revisited their data analyses, which are now better supported by the statistical analyses and conceptualization.

Minor point.

At line 66 and 916 it is written "When formulating therapeutic strategies that involve fast-spiking neurons, it is crucial to take into account the molecular and functional species differences." This assertion could benefit from clearer context or citations to avoid sounding speculative or overselling the research findings.

---

## [Editor Report · Decision Letter 4]

24 Nov 2025

Dear Dr Lamsa,

Thank you for the submission of your revised Research Article "Adaptations of the axon initial segment in fast-spiking interneurons of the human neocortex support low action potential thresholds" for publication in PLOS Biology. On behalf of my colleagues and the Academic Editor, Alberto Bacci, I am pleased to say that we can in principle accept your manuscript for publication, provided you address any remaining formatting and reporting issues. These will be detailed in an email you should receive within 2-3 business days from our colleagues in the journal operations team; no action is required from you until then. Please note that we will not be able to formally accept your manuscript and schedule it for publication until you have completed any requested changes.

PRESS

Sincerely, 

Taylor

Taylor Hart, PhD,

Associate Editor

PLOS Biology

thart@plos.org